# A co-ordinated transcriptional programme in the maternal liver supplies long chain polyunsaturated fatty acids to the conceptus using phospholipids

Risha Amarsi [1,2], Samuel Furse [3,4], Mary A. M. Cleaton [5], Sarah Maurel[6], Alice L. Mitchell [7], Anne C. Ferguson-Smith[5], Nicolas Cenac [6], Catherine Williamson [7], Albert Koulman [4] & Marika Charalambous [1,2] ✉

The long and very long chain polyunsaturated fatty acids (LC-PUFAs) are preferentially transported by the mother to the fetus. Failure to supply LC-PUFAs is strongly linked with stillbirth, fetal growth restriction, and impaired neurodevelopmental outcomes. However, dietary supplementation during pregnancy is unable to simply reverse these outcomes, suggesting imperfectly understood interactions between dietary fatty acid intake and the molecular mechanisms of maternal supply. Here we employ a comprehensive approach combining untargeted and targeted lipidomics with transcriptional profiling of maternal and fetal tissues in mouse pregnancy. Comparison of wild-type mice with genetic models of impaired lipid metabolism allows us to describe maternal hepatic adaptations required to provide LC-PUFAs to the developing fetus. A late pregnancy-specific, selective activation of the Liver X Receptor signalling pathway dramatically increases maternal supply of LC-PUFAs within circulating phospholipids. Crucially, genetic ablation of this pathway in the mother reduces LC-PUFA accumulation by the fetus, specifically of docosahexaenoic acid (DHA), a critical nutrient for brain development.

During a normal pregnancy, maternal lipid metabolism undergoes striking adaptations to meet the nutritional demands of the developing fetus. Anabolic pathways in early pregnancy promote the net accumulation of nutrients within maternal tissues, primarily in the form of triglycerides[1]. A mid-gestation catabolic switch then drives the breakdown of nutrient stores, ensuring the circulating availability of fatty acids and glucose for fetal uptake. These dynamically regulated shifts in maternal nutrient allocation are vital for a healthy pregnancy,

as poor adaptations, commonly seen in pregnancies with obesity, can profoundly disrupt the development and long-term metabolic health of the child[2].

Rodent studies have helped to characterise the anabolic and catabolic shifts in whole-body lipid metabolism[3]. In a previous study we demonstrated that maternal plasma Delta-like homologue 1 (DLK1) levels are elevated in the catabolic phase of pregnancy, and that the major source of this protein was the conceptus[4]. *Dlk1* is an imprinted

[1]Department of Medical and Molecular Genetics, Faculty of Life Sciences and Medicine, King's College London, London SE19RT, UK. [2]Pregnancy Physiology Laboratory, Francis Crick Institute, 1 Midland Road, NW1 1AT London, UK. [3]Biological chemistry group, Jodrell laboratory, Royal Botanic Gardens Kew, Kew Road, Richmond, Surrey TW9 3DS, UK. [4]Core Metabolomics and Lipidomics Laboratory, Wellcome-MRC Institute of Metabolic Science, University of Cambridge, Addenbrooke's Treatment Centre, Keith Day Road, Cambridge CB2 0QQ, UK. [5]Department of Genetics, Downing Street, University of Cambridge, Cambridge CB2 3EH, UK. [6]IRSD, Université de Toulouse-Paul Sabatier, INSERM, INRAe, ENVT, UPS, Toulouse, France. [7]Department of Women and Children's Health, King's College London, Guy's Campus, London, UK. ✉e-mail: marika.charalambous@kcl.ac.uk

**Fig. 1 | Study design. A** We used liver and plasma samples from a previously published cohort with manipulations to the imprinted gene, *Dlk1*. Numbers within the boxes represent group number and previously reported maternal plasma DLK1 concentration at 15.5 dpc (italic, in ng/mL[4]). The study included five pregnant groups at 15.5 dpc (groups 2, 4, 5, 7 and 8) and three age and genotype-matched virgin groups (groups 1, 3, and 6). Groups 1 and 2 were genetically unmodified females with normal DLK1 expression. Groups 6, 8 and 7 inherited a silent *Dlk1* deletion from their mother and so maintained a functional copy of DLK1. Group 6 virgins replicate group 1 virgins, and group 8 replicate group 2 since they have normal conceptus-derived plasma DLK1 expression. Group 7 females were crossed to a *Dlk1*⁻/⁻ sire, thus lack conceptus-derived plasma DLK1 expression. Groups 3, 5 and 4 were *Dlk1*⁻/⁻ females and so did not have a functional copy of DLK1. Group 5 females had normal conceptus-derived plasma DLK1 expression while group 4 females were crossed to a *Dlk1*⁻/⁻ sire and so lacked conceptus-derived plasma DLK1. **B** To test the effect of normal pregnancy on liver and plasma lipids, we conducted three replicate comparisons between virgin and pregnant groups of matched maternal *Dlk1* genotype. **C** The influence of conceptus-derived circulating DLK1 on the normal pregnant lipidome was then investigated by two replicate comparisons between pregnant groups with and without a functional fetal *Dlk1* copy in matched genotype dams ("Fetal Effect"). Similarly, the influence of DLK1 derived from maternal tissues on the normal pregnant lipidome was tested by comparing DLK1+ with DLK1- dams with matched fetal DLK1 production ("Maternal Effect").

gene expressed predominantly from the paternally-inherited allele[5,6]. This mode of inheritance allowed us to independently examine the influence of *Dlk1* loss of function intrinsically in maternal tissues or as an endocrine factor produced by the fetus. We observed that loss of DLK1 in the fetus caused impairments in maternal fasting metabolism and lipoprotein production, whereas loss of DLK1 in the dam prevented the normal acquisition and release of her adipose tissue stores[4]. This work led us to hypothesise that DLK1 is a key modulator of maternal fatty acid metabolism in pregnancy, yet a detailed examination of pregnancy-associated lipid species has not been performed.

Fatty acids (FAs) are crucial components at every stage of fetal development, both as an energy source and as building blocks and key signals for organogenesis[7]. Omega 6 (n-6) and omega 3 (n-3) LC-PUFAs such as arachidonic acid (ARA, 20:4n-6), eicosapentaenoic acid (EPA, 20:5n-3) and docosahexaenoic acid (DHA, 22:6n-3), are preferentially transported by the mother to the fetus, a process known as biomagnification[8]. Both n-6 and n-3 FAs are essential for fetal development, with roles in placentation, synthesis of membrane components and membrane fluidity, and the production of signalling molecules[9]. They are especially important for brain growth and central nervous system development, since this organ contains 50-60% dry mass as lipid, ~35% of which are LC-PUFAs[10]. Fetal synthesis can only account for a small proportion of this demand, and instead maternal production/mobilisation and placental transfer is required to meet the ARA and DHA requirements for healthy development[8,11]. Precursors of LC-PUFAs, linoleic acid (LA, 18:2n-6) and alpha-linolenic acid (ALA, 18:3n-3), can only be obtained from dietary sources. Through elongation/desaturation reactions, LA is used to synthesise omega-6 (n-6) and ALA is converted into omega-3 (n-3) LC-PUFAs, although only a small amount of nutrient requirement is thought to be met by this route in the non-pregnant state[12]. The mechanism driving the biomagnification of LC-PUFAs between maternal and fetal compartments is not fully understood.

Although circulating adipose-derived non-esterified fatty acids (NEFAs) can be directly transported across the placenta, the majority are transferred to the maternal liver, a highly adaptive, yet poorly characterised metabolic tissue in pregnancy. Hepatic re-esterification of fatty acids into triglycerides and export into the circulation within very low-density lipoproteins (VLDL) underlies the characteristic rise in plasma triglycerides in late gestation and is considered the primary source of fatty acids for the fetus[7]. In normal physiology, the liver modulates complex synthetic and catabolic pathways to regulate whole-body lipid dynamics. Since genetic polymorphisms within

known hepatic lipid pathway genes drive at least 10% of population variation in plasma triglycerides[13,14], we propose that hepatic pathways are likely regulators of fatty acid allocation in pregnancy.

However, because of the focus on hepatic triglycerides as the main fetal source of LC-PUFAs, unbiased mechanistic investigations into lipid synthetic pathways in the maternal liver are scarce. In recent years, lipidomic investigations of pregnancy have provided a molecular insight into the fatty acid composition of lipids, with numerous phospholipid biomarkers reported for common metabolic complications of pregnancy[15–17]. Phospholipids are well-reported to rise in the maternal circulation[18] and could represent a substantial physiological source of LC-PUFAs in plasma and tissues[19–21]. These studies suggested to us that biomagnification pathways may extend beyond production of maternal circulating triglycerides and involve LC-PUFA incorporation into other types of lipid.

Here we profiled the liver and plasma lipidomes of virgin and pregnant mice as an untargeted discovery approach for lipid pathways in the maternal liver that are associated with maternal fatty acid provision to the conceptus. With the aid of our *Dlk1* model of disrupted lipid metabolism and targeted PUFA metabolite and transcriptional profiles, we aimed to delineate the major hepatic lipid pathways that underlie the fatty acid adaptations of late pregnancy. We show that i) ARA and DHA are enriched in the maternal liver and in the circulation selectively in the form of phospholipids; ii) Hepatic transcriptional pathways that synthesise LC-PUFAs from dietary intermediates and incorporate them into phospholipids are activated in late pregnancy; iii) This transcriptional programme is co-ordinately regulated by the liver X receptor (LXR) and modulated by maternal production of DLK1. Taken together we propose that biomagnification is achieved by a co-ordinated regulatory programme in the mother to supply LC-PUFAs to the conceptus in the phospholipid compartment.

## Results

In the current study we used samples from a previously published cohort to investigate both maternal lipid metabolism in normal pregnancy, and perturbations to this process as a result of loss of DLK1[4]. Liver and plasma were collected from matched virgin and pregnant dams at 15.5 dpc in 8 groups with modified DLK1 in either the maternal tissue or in the plasma (derived from the conceptus), Fig. 1A. We hypothesised that i) the abundance of specific lipid species would be co-ordinately modified in late pregnancy in the plasma and liver. To test this we compared 3 replicate groups of matched maternal genotype (Fig. 1B). ii) Circulating DLK1 generated by the conceptus can

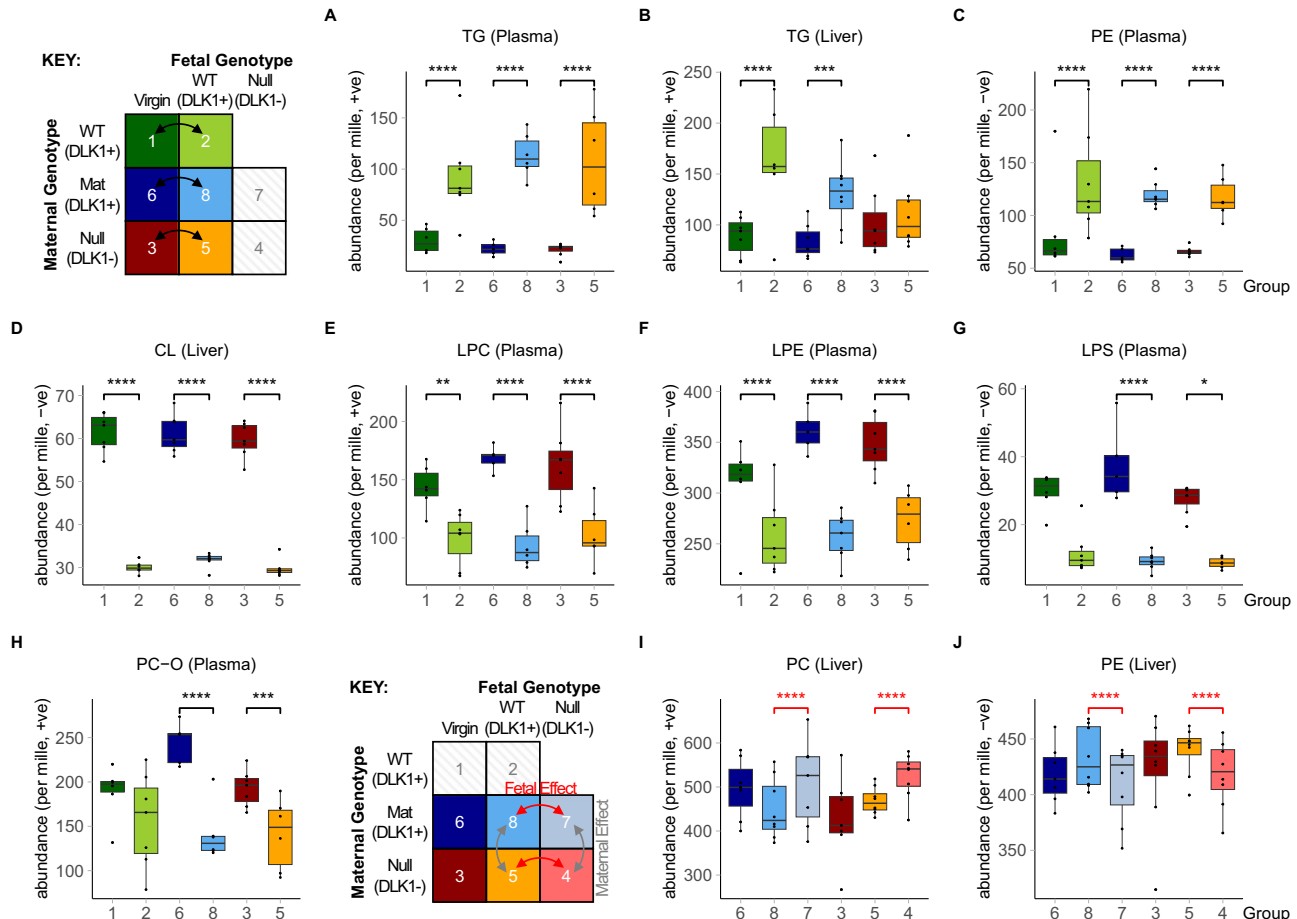

**Fig. 2 | Lipid classes that change in the liver or plasma of mice in normal and *Dlk1*-manipulated pregnancy. A–J** Grouped relative abundance of lipid classes that are significantly different in liver or plasma in pregnant mice (15.5 dpc) compared to virgin controls (**A–H**), and in pregnant mice that lack fetal or maternal-derived DLK1 protein (**I–J**). Significance was only considered if identified in at least two genotype-matched replicate group comparisons. All class data are found in Supplementary Data Tables S1–2. Data is presented as boxplots (median and IQR (25th and 75th percentiles) with whiskers showing 1.5*IQR) with individual values. Two-way ANOVA with Sidak's multiple comparisons test was performed to determine significant class shifts between experimental groups (*p-value < 0.05; **p-value < 0.01; ***p-value < 0.001; ****p-value < 0.0001). Statistical tests were performed independently per ionisation mode and per genotype-matched replicate comparison. p-values for A: 1vs2 = $1.60 \times 10^{-05}$, 6vs8 = $1.00 \times 10^{-15}$, 3vs5 = $8.36 \times 10^{-10}$; B: 1vs2 = $1.20 \times 10^{-07}$, 6vs8 = 0.0002, 3vs5 = NS; C: 1vs2 = $6.54 \times 10^{-05}$,

6vs8 = $1.00 \times 10^{-15}$, 3vs5 = $5.00 \times 10^{-15}$; D: 1vs2 = $7.50 \times 10^{-14}$, 6vs8 = $8.34 \times 10^{-10}$, 3vs5 = $8.70 \times 10^{-14}$; E: 1vs2 = 0.0074, 6vs8 = $2.01 \times 10^{-12}$, 3vs5 = $5.82 \times 10^{-05}$; F: 1vs2 = $7.43 \times 10^{-06}$, 6vs8 = $1.00 \times 10^{-15}$, 3vs5 = $1.00 \times 10^{-15}$; G: 1vs2 = NS, 6vs8 = $1.54 \times 10^{-06}$, 3vs5 = 0.028; H: 1vs2 = NS, 6vs8 = $1.00 \times 10^{-15}$, 3vs5 = 0.0009; I: 8vs7 = $8.39 \times 10^{-05}$, 5vs4 = $3.00 \times 10^{-15}$; J: 8vs7 = $2.61 \times 10^{-05}$, 5vs4 = $5.82 \times 10^{-07}$. Plasma data: n = 6 (group 1), n = 7 (group 2), n = 7 (group 3), n = 5 (group 4; +ve mode) n = 8 (group 4; −ve mode), n = 6 (group 5), n = 5 (group 6), n = 8 (group 7), n = 6 (group 8; +ve mode), n = 7 (group 8; −ve mode); Liver data: n = 7 (group 1), n = 6 (group 2), n = 7 (group 3), n = 8 (group 4), n = 8 (group 5), n = 7 (group 6), n = 7 (group 7, +ve mode), n = 8 (group 7; −ve mode), n = 8 (group 8); mice per group. CL cardiolipin, LPC *lyso*-phosphatidylcholine, LPE *lyso*-phosphatidylethanolamine, LPS *lyso*-phosphatidylserine, PC phosphatidylcholine, PC-O PC plasmalogen, PE phosphatidylethanolamine, TG triglyceride. Source data are provided as a Source Data file.

influence maternal plasma-liver lipid abundance. Here we compared groups with and without a functional fetal *Dlk1* copy in matched genotype dams (Fig. 1C, 'Fetal Effect'. iii) Finally, we hypothesised that maternal *Dlk1* genotype would influence the abundance of maternal lipids, and tested this by comparing dams with matched fetal DLK1 production (Fig. 1C, 'Maternal Effect').

### The maternal circulating and hepatic lipid profiles undergo broad changes to triglycerides and phospholipids in late pregnancy

We performed an untargeted lipidomics screen of plasma and liver samples from the cohort described above, utilising Direct Infusion high-resolution Mass Spectrometry (DI-MS, see Materials and Methods) and classified the resulting lipidomics dataset into major lipid classes, comparing their relative abundance between the three genotype-matched virgin and pregnant group-pairs (Supplementary

Data Tables S1, 2). As expected in the catabolic phase of pregnancy, triglycerides (TG) were considerably higher in plasma and liver (Fig. 2A, B). By contrast, the most abundant lipid class, phosphatidylcholine (PC), did not show consistent changes in liver or plasma as a result of pregnancy (Supplementary Fig. S1A–D). Similarly, phosphatidylethanolamine (PE), a major membrane phospholipid, was only increased in plasma, and was unchanged in the livers of pregnant mice (Fig. 2C, Supplementary Fig. S1E). We observed shifts in less abundant phospholipids; cardiolipins (CL), which constitute the principal lipid component of mitochondria, were reduced in the livers of pregnant groups (Fig. 2D). Moreover, the *lyso*-phospholipids (LPC, LPE and LPS), which are formed by the hydrolysis of one fatty acid residue from a phospholipid, as well as plasmalogens (ether-linked lipids, e.g. PC-O), were all decreased in the plasma of pregnant mice when compared to virgin females (Fig. 2E–H). The similarities in abundance between conditions for hepatic PC and PE, and the reduced hepatic CL in

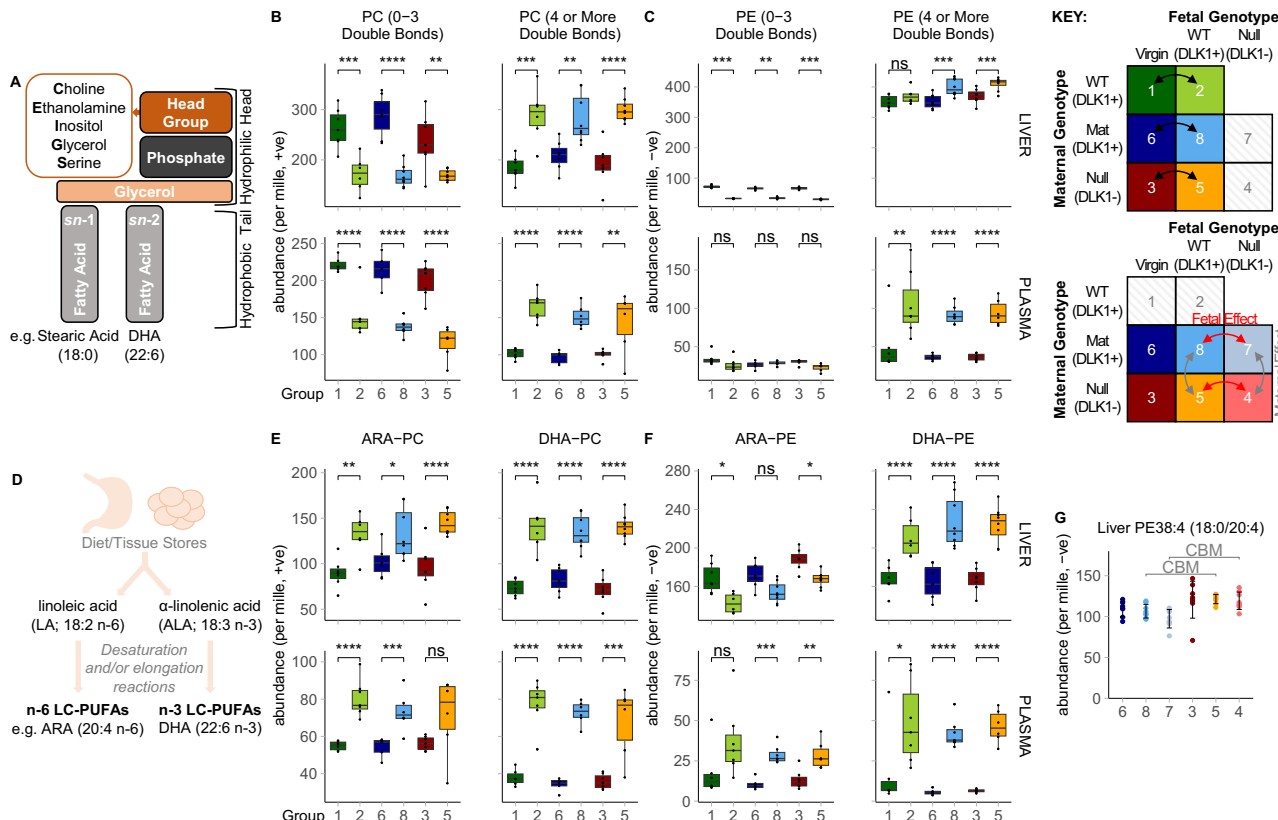

**Fig. 3 | Changes to the fatty acid composition of PC and PE lipids in normal and *Dlk1*-manipulated pregnancy. A** Phospholipid structure. Phospholipids contain various head groups and are commonly associated with a saturated or mono-unsaturated fatty acid at the *sn*−1 position and a monounsaturated or poly-unsaturated fatty acid at the *sn*−2 position. **B, C** Grouped relative abundance of PC (**B**) and PE (**C**) lipids that contain fatty acids with a combined total of three or fewer double bonds (left) or four or more double bonds (right) in the liver and plasma of virgin and pregnant (15.5 dpc) groups. PC data in the negative ionisation mode is shown in Supplementary Fig. S2A. **D** Schematic of n-6 and n-3 LC-PUFA synthesis from essential fatty acid precursors. **E, F** Grouped relative abundance of the most common PC (**E**) and PE (**F**) lipids that specifically contain ARA (left) or DHA (right), as identified by targeted LC-MS/MS analysis. List of included lipids are found in Supplementary Data Table S7. Grouped abundance data (**B, C, E, F**) is presented as boxplots (median and IQR (25th and 75th percentiles) with whiskers showing 1.5*IQR) with individual values. Each genotype-matched virgin vs pregnant comparison in (**B, C, E, F**) was compared by two-way ANOVA with Sidak's multiple comparisons (*p*-value < 0.05; **p*-value < 0.01; ***p*-value < 0.001; ****p*-value < 0.0001). Significance was only considered if identified in at least two genotype-matched replicate comparisons. p-values for (**B**) - Liver PC (0-3 Double Bonds): 1vs2 = 0.0007, 6vs8 = 1.88 × 10⁻⁰⁶, 3vs5 = 0.002; (**B**) - Liver PC (4+ Double Bonds): 1vs2 = 0.0001, 6vs8 = 0.0012, 3vs5 = 6.62 × 10⁻⁰⁶; (**B**) - Plasma PC (0-3 Double Bonds): 1vs2 = 2.84 × 10⁻⁰⁶, 6vs8 = 2.58 × 10⁰⁷, 3vs5 = 9.10 × 10⁻⁰⁶; (**B**) - Plasma PC (4+ Double Bonds): 1vs2 = 1.04 × 10⁻⁰⁵, 6vs8 = 2.38 × 10⁻⁰⁵, 3vs5 = 0.0096; (**C**) - Liver PE (0-3 Double Bonds): 1vs2 = 0.0007, 6vs8 = 0.0059, 3vs5 = 0.0004; (**C**) - Liver PE (4+ Double Bonds): 1vs2 = NS, 6vs8 = 0.0001, 3vs5 = 0.0002; C - Plasma PE (0-3 Double Bonds): 1vs2 = NS, 6vs8 = NS, 3vs5 = NS; C - Plasma PE (4+ Double Bonds): 1vs2 =

0.0064, 6vs8 = 1.18 × 10⁻¹⁰, 3vs5 = 1.07 × 10⁻¹⁰. *p*-values for (**E**) - Liver ARA-PC: 1vs2 = 0.0016, 6vs8 = 0.014, 3vs5 = 1.82 × 10⁻⁰⁵; (**E**) - Liver DHA-PC: 1vs2 = 6.59 × 10⁻⁰⁶, 6vs8 = 5.99 × 10⁻⁰⁵, 3vs5 = 5.96 × 10⁻⁰⁸; (**E**) - Plasma ARA-PC: 1vs2 = 4.00 × 10⁻⁰⁵, 6vs8 = 0.0006, 3vs5 = NS; E - Plasma DHA-PC: 1vs2 = 5.17 × 10⁻⁰⁸, 6vs8 = 8.44 × 10⁻⁰⁸, 3vs5 = 0.0006; (**F**) - Liver ARA-PE: 1vs2 = 0.011, 6vs8 = NS, 3vs5 = 0.012; F - Liver DHA-PE: 1vs2 = 7.78 × 10⁻⁰⁵, 6vs8 = 9.98 × 10⁻⁰⁷, 3vs5 = 1.78 × 10⁻⁰⁸; (**F**) - Plasma ARA-PE: 1vs2 = NS, 6vs8 = 0.0001, 3vs5 = 0.0017; (**F**) - Plasma DHA-PE: 1vs2 = 0.043, 6vs8 = 4.85×10⁻⁰⁹, 3vs5 = 2.02×10⁻⁰⁹. Plasma data: *n* = 6 (group 1), *n* = 7 (group 2), *n* = 7 (group 3), *n* = 6 (group 5), *n* = 5 (group 6), *n* = 6 (group 8; +ve mode), *n* = 7 (group 8; −ve mode); Liver data: *n* = 7 (group 1), *n* = 6 (group 2), *n* = 7 (group 3), *n* = 8 (group 5), *n* = 7 (group 6), *n* = 8 (group 8); mice per group. **G** Relative abundance of PE38:4 which was identified as a candidate bio-marker (CBM) that distinguishes the livers of pregnant dams lacking maternal-derived DLK1 from those with normal expression of maternal-derived DLK1. CBMs are classified as lipids that passed both Bonferroni-adjusted two-tailed *t*-tests (liver threshold, p = 0.00234) and sparse partial least squares discriminant analysis in two genotype-matched replicate comparisons (see Supplementary Data Table S5 for full list of CBMs). CBM data is shown as individual relative abundance values with ± SD error bars and *sn*−1/*sn*−2 fatty acid compositions were assigned by targeted LC-MS/MS analysis (Supplementary Data Table S6). *n* = 8 (group 3), *n* = 8 (group 4), *n* = 8 (group 5), *n* = 7 (group 6), *n* = 8 (group 7), *n* = 8 (group 8); mice per group. All statistical tests were performed independently per ionisation mode and per genotype-matched replicate comparison. ARA arachidonic acid, DHA docosahex-aenoic acid, PC phosphatidylcholine, PE phosphatidylethanolamine. Source data are provided as a Source Data file.

pregnant groups were supported by phosphorus NMR profiling of pooled liver samples (Supplementary Data Table S3).

Since class-wide changes in hepatic PC and PE were not evident when comparing pregnant and virgin groups, we investigated whether maternal phospholipids were altered based on their fatty acid compositions. Phospholipids contain fatty acid chains in two distinct bio-chemical positions on the glycerol moiety; the *sn*-1 position is typically occupied by a saturated fatty acid whereas the *sn*-2 position is more often occupied by an unsaturated residue. DI-MS is unable to

distinguish which individual fatty acids occupy these positions, but rather reports the sum of carbon chains and unsaturated bonds on both positions (e.g. 40:6 in the example in Fig. 3A). We grouped PC lipids by the total number of double bonds on their fatty acid chains and found that those with four or more double bonds were increased while those with three or fewer were decreased in pregnant groups (Fig. 3B, Supplementary Fig. S2A). Thus, we found a substantial shift in abundance between subclasses of PC. This unsaturation-specific directionality was apparent in both liver and plasma. A similar

**Table 1 | PC and PE species of interest that were identified as candidate biomarkers associated with pregnancy**

| Saturation (# double bonds) | Lipid | Fatty Acid Composition (isoform abundance) | Compartment | CBM (Y/N) | Change in Pregnancy |
|---|---|---|---|---|---|
| 4 or more | PC(37:4) | 17:0/20:4 (% undetermined) | Plasma | Y | UP |
| | | 17:0/20:4 (% undetermined) | Liver | Y | UP |
| 4 or more | PC(38:6) | 16:0/22:6 (85%) | Plasma | Y | UP |
| | | 16:0/22:6 (66%) | Liver | Y | UP |
| 4 or more | PC(39:6) | 17:0/22:6 (60%) | Plasma | Y | UP |
| | | 17:0/22:6 (85%) | Liver | Y | UP |
| 4 or more | PC(40:6) | 18:0/22:6 (95%) | Plasma | Y | UP |
| | | 18:0/22:6 (95%) | Liver | Y | UP |
| 4 or more | PE(40:6) | 18:0/22:6 (85%) | Plasma | Y | UP |
| | | 18:0/22:6 (85%) | Liver | Y | UP |
| 3 or less | PE(34:2) | 16:0/18:2 (100%) | Plasma | N | n/a |
| | | 16:0/18:2 (100%) | Liver | Y | DOWN |
| 3 or less | PE(36:2) | 18:0/18:2 (85%) | Plasma | N | n/a |
| | | 18:0/18:2 (75%) | Liver | Y | DOWN |
| 3 or less | PC(34:2) | 16:0/18:2 (95%) | Plasma | Y | DOWN |
| | | 16:0/18:2 (95%) | Liver | N | n/a (trending down) |
| 3 or less | PC(36:3) | 18:1/18:2 (55%) | Plasma | Y | DOWN |
| | | 18:1/18:2 (50%); 16:0/20:3 (50%) | Liver | Y | DOWN |
| 3 or less | PE(34:1) | did not analyse | Plasma | N | n/a |
| | | did not analyse | Liver | Y | DOWN |

PC and PE candidate biomarkers (CBMs) that distinguish pregnant groups (15.5 dpc; groups 2, 8 and 5) from virgin groups (groups 1, 6 and 3) were selected based on their level of unsaturation and the tissue compartment in which they were identified. *sn*–1/*sn*–2 fatty acid compositions were assigned using the most abundant isoform identified from targeted LC-MS/MS analysis in plasma and liver (percentage of total lipid/glyceride signal indicated in brackets; see Supplementary Data Table S6). CBMs were classified as lipids that passed both Bonferroni-adjusted two-tailed *t*-tests (liver threshold, *p* = 0.00234; plasma threshold, p = 0.00283) and sparse partial least squares discriminant analysis in at least two genotype-matched virgin vs pregnant comparisons (see Supplementary Data Table S4 for full CBM list). Individual CBM plots are depicted in Supplementary Fig. S3. All CBM tests were performed independently per ionisation mode and per genotype-matched replicate comparison. Plasma data: *n* = 7 (group 1), *n* = 7 (group 2), *n* = 7 (group 3), *n* = 6 (group 5), *n* = 5 (group 6), *n* = 7 (group 8); Liver data: *n* = 8 (group 1; +ve mode), *n* = 7 (group 1; –ve mode), *n* = 7 (group 2), *n* = 7 (group 3; +ve mode), *n* = 8 (group 3; –ve mode), *n* = 8 (group 5), *n* = 7 (group 6), *n* = 8 (group 8). *PC* phosphatidylcholine, *PE* phosphatidylethanolamine. Source data are provided as a Source Data file.

abundance shift was evident for PE lipids in liver (Fig. 3C). In plasma, where PEs are much less abundant, PEs with four or more double bonds increased in abundance, however, those with three or fewer were unchanged in pregnancy. Hence, while triglycerides show a characteristic class-wide increase in the livers and circulation of pregnant mice, the two major phospholipid classes ( ~ 80-90% of total phospholipid) instead undergo selective shifts to their fatty acid composition in favour of a more unsaturated phenotype, suggesting pregnancy-associated changes to the control of the system.

We next evaluated the lipid class profile of pregnancy using our *Dlk1*-model of perturbed maternal lipid adaptations. While triglycerides and less abundant phospholipid classes were unchanged between DLK1+ and DLK1- pregnant groups, we observed a shift in the most abundant phospholipid classes in the maternal liver (Supplementary Data Table S2). Specifically, pregnant dams that lacked fetal-derived circulating DLK1 had higher amounts of PC and reduced amounts of PE in their livers, compared to those with normal circulating DLK1 (Fig. 2I, J). Unlike the saturation-specific effect of normal pregnancy, the DLK1-dependent shift in abundance of these phospholipids was due to small alterations to several lipids within the class (Supplementary Fig. S2B–D), suggesting that DLK1 derived from fetal tissues has a class-wide influence on maternal hepatic PC and PE abundance in pregnancy.

**Circulating ARA and DHA in late pregnancy is driven by the selective hepatic production and export of phospholipids, not triglycerides**

To identify the molecular lipid species that best distinguished pregnant from virgin groups we next applied a stringent statistical pipeline, known as "Candidate Biomarker" (CBM) discovery[22]. In at least two replicate virgin vs pregnant group comparisons, lipids that passed the

sparse Partial Least Squares Discriminant Analysis (sPLS-DA) multi-variate test, followed by an FDR-corrected Student's *t*-test, were classified as CBMs. Of the resulting 43 CBMs, 29 were PCs or PEs, suggesting that the observed shifts in maternal phospholipids are represented by select PC and PE species (Supplementary Data Table S4).

We used targeted LC-MS/MS to quantify the FA composition of PC and PE CBMs that were driving the unsaturation-specific shifts in maternal phospholipids (Supplementary Data Table S6). Notably, those that contained four or more double bonds, PC(37:4, 38:6, 39:6, 40:6) and PE(40:6), were all raised in both liver and plasma, and contained DHA or ARA (Table 1; Supplementary Fig. S3A). These data reveal a selective rise in phospholipids that contain the two most biologically important LC-PUFAs in pregnancy. By contrast, PCs that contained three or fewer double bonds, PC(34:2, 35:3, 36:3), were lower in pregnant groups and contained LA (18:2 n-6), the diet-derived precursor fatty acid in n-6 LC-PUFA synthesis (Table 1; Supplementary Fig. S3B; Fig. 3D). The specific alterations in the profile of PUFAs in PC suggests that the hepatic LC-PUFA pathway and their incorporation into phospholipids are key regulatory targets in late pregnancy. Moreover, the parallel changes in liver and plasma implicate the hepatic transfer of these particular lipids into the circulation.

Pregnancy-associated CBMs were evaluated further by comparison to the most abundant ARA or DHA-containing PCs and PEs of the mouse liver using a published lipidomics profile[19] and targeted LC-MS/MS to identify and quantify the FAs they comprised (list of included lipids in Supplementary Data Table S7). When the combined abundances were compared between virgin and pregnant groups, ARA-PC, DHA-PC and DHA-PE were considerably more abundant in both maternal liver and plasma, the PC lipids showing concurrence between the two compartments (Fig. 3E, F). These data highlight the

**Table 2 | Relative abundance of PC and triglycerides that contain a LC-PUFA in the liver and plasma of virgin and pregnant mice**

| Compartment | Class | Relative Abundance (mean ± sd, per mille) | | | |
| | | LC-PUFA-containing | | Non-LC-PUFA-containing | |
| | | VIRGIN | PREGNANT | VIRGIN | PREGNANT |
|---|---|---|---|---|---|
| Plasma | PC | 99.66 (6.93) | 154.13 (25.94) | 211.09 (19.65) | 135.9 (25.65) |
| Plasma | TG | 1.19 (1.40) | 24.21 (9.50) | 23.33 (7.99) | 79.8 (29.81) |
| Plasma | DG | 1.46 (3.40) | 7.19 (2.74) | 133.79 (37.22) | 183.6 (51.03) |
| Liver | PC | 194.52 (31.5) | 290.5 (39.93) | 262.68 (46.82) | 169.41 (21.71) |
| Liver | TG | 11.5 (5.04) | 19.39 (6.28) | 80.37 (21.60) | 112.6 (38.13) |
| Liver | DG | 33.25 (5.96) | 38.93 (8.19) | 208.79 (33.19) | 201.5 (32.53) |

The relative abundance of PC, triglycerides and diglycerides (which represent fragmented triglycerides) were grouped based on whether they contain a LC-PUFA. LC-PUFA-containing PCs were assigned based on targeted LC-MS/MS analysis (Supplementary Data Table S6). Triglyceride signals with 54 or more carbons and 5 or more double-bonds, and diglyceride signals with 38 or more carbons and 5 or more double-bonds, were expected to contain a LC-PUFA. Data is presented in the positive ionisation mode to highlight the proportion of LC-PUFAs in triglycerides compared to PC. Data is combined for pregnant (15.5 dpc; groups 2, 8 and 5) and virgin (groups 1, 6 and 3) groups and presented as mean relative abundance (± SD). Plasma data: $n = 7$ (group 1), $n = 7$ (group 2), $n = 7$ (group 3), $n = 6$ (group 5), $n = 5$ (group 6), $n = 7$ (group 8); Liver data: $n = 8$ (group 1). $n = 7$ (group 2), $n = 7$ (group 3), $n = 8$ (group 5), $n = 7$ (group 6), $n = 8$ (group 8). DG diglyceride, LC-PUFA long-chain polyunsaturated fatty acid, PC phosphatidylcholine, PE phosphatidylethanolamine, TG triglyceride. Source data are provided as a Source Data file.

wide-spread increase of LC-PUFA-containing phospholipids in late pregnancy, and suggest that the maternal liver selectively promotes the generation and export of phospholipids that contain ARA and DHA.

Although triglycerides are considered a major supply route for LC-PUFAs to the fetus, only one minor triglyceride species, TG(56:6), was identified as a CBM that was different between virgin and pregnant groups (Supplementary Data Table S4). Hence, in contrast to phospholipids, the observed class-wide rise in plasma and hepatic triglycerides is interpreted as the sum of minor increases by each individual triglyceride. As triglycerides and phospholipids have very different ionisation efficiencies and it is therefore difficult to compare concurrent changes in both groups, it is sometimes mores useful to determine within group changes. Indeed, when we split triglycerides into those that do and those that do not contain LC-PUFAs, both types were raised in pregnant groups compared to virgins (Table 2). Crucially, data in Table 2 highlights how the proportional abundance of LC-PUFAs is considerably higher in PCs compared to triglycerides and diglycerides. We therefore propose that the general rise in maternal triglycerides increases the supply of all fatty acids to the fetus, while the more regulated shifts in PC composition substantially increases the availability of LC-PUFAs for fetal uptake.

## The n-6 PUFA composition of PEs undergo liver-specific modifications in late pregnancy

An exception to the rise in hepatic and circulating LC-PUFA-phospholipids in late pregnancy was observed for ARA-containing PE, which was increased in plasma but reduced in livers of pregnant mice (Fig. 3F). We tested whether the shifts in individual PE species were specific to one of the two maternal compartments. With the exception of the highly abundant DHA-containing lipid, PE(40:6), which increased in both liver and plasma, all other PE CBMs were only identified in one compartment (Supplementary Data Table S4). This is in direct contrast to the parallel changes to nearly all PC CBMs.

Two abundant PE lipids that contain LA (18:2 n-6), PE(34:2, 36:2), were lower in pregnant groups, but unlike LA-containing PC CBMs, this fall was specific to the maternal liver (Table 1, Supplementary Fig. S3B). Of the three most abundant ARA-containing PE species, PE(38:5) was lower only in the livers of pregnant mice, while PE(38:4, 36:4), were higher, but only in plasma. Of note, around 20% of PE(38:5) contains EPA (20:5 n-3), rather than ARA, as determined on the basis of fragmentation data obtained by targeted LC-MS/MS (Supplementary Data Table S6). These data together suggest a species-specific regulation of ARA-PE and EPA-PE in the liver, and warrants further investigation into isoform-specific adaptations to hepatic PE in pregnancy.

When we used the CBM analysis to compare individual lipids between DLK1+ and DLK1- pregnant groups, PE(38:4) was identified as a CBM that was increased in the livers of pregnant mice that lacked DLK1 in maternal tissues (Fig. 3G, Supplementary Data Table S5). This lipid is notably the most abundant of all isoforms of PE, and thus changes in it represent not only a clear shift in the configuration of the system but also a sizable proportion of hepatic ARA. Taken together, these data reveal the complex control of n-6 PUFAs in PE in the maternal liver, which may in part be influenced by maternally-produced DLK1.

## ARA-containing phospholipids are hydroxylated in the maternal liver to generate bioactive lipids of the cyclooxygenase pathway

The enzymatic cleavage of PUFAs, principally ARA, from membrane phospholipids is the rate-limiting step in in the generation of bioactive PUFA metabolites[23]. To investigate whether the maternal liver uses ARA-phospholipids to enhance their biosynthesis, a PUFA metabolite panel was run on the virgin and pregnant liver samples. LC-MS/MS was used to measure the concentrations of 38 metabolites that are synthesised from PUFAs via cyclooxygenase (COX), lipoxygenase (LOX), Cytochrome P450 (CYP450) or non-enzymatic beta-oxidation pathways (Fig. 4A). To identify metabolites that best distinguished pregnant from virgin groups, we applied the CBM discovery statistical technique, which identified four COX-synthesised metabolites (prostaglandin D2, prostaglandin E2, prostaglandin F2α and thromboxane B2) and two LOX-synthesised metabolites (5-hydroxyeicosatetraenoic acid and lipoxin A4) as CBMs, all of which were higher in the maternal liver (Fig. 4B−H). Immunohistochemistry on sections from virgin and pregnant livers confirmed the hepatic expression of the COX-1 protein, which was located within and in close proximity to endothelial cells (Fig. 4I, J). Notably, all six CBMs were derived from ARA, which mirrors the generally higher levels of ARA-derived metabolites observed in the COX, LOX and CYP450 pathways (Fig. 4A). Despite the influence of DLK1 on hepatic PE(18:0/20:4), no effect was observed on PUFA metabolite concentration (Supplementary Data Table S8). Taken together, the reduced hepatic abundance of ARA-containing PE in pregnancy may be a result of increased PUFA metabolite biosynthesis in the late pregnant liver, particularly within the cyclooxygenase pathway.

## Hepatic biosynthesis and export of LC-PUFA-phospholipids is transcriptionally upregulated in pregnancy

We speculated that enzymatic pathways involved in the synthesis and export of LC-PUFA-phospholipids were crucial targets for adaptation of the late gestational liver. To investigate candidate pathways, we re-analysed a microarray dataset that compared liver transcriptomes between virgin and pregnant mice at 14.5 dpc[24] (Fig. 5A), which resulted in 3272 differentially expressed genes (Fig. 5B, C; Supplementary

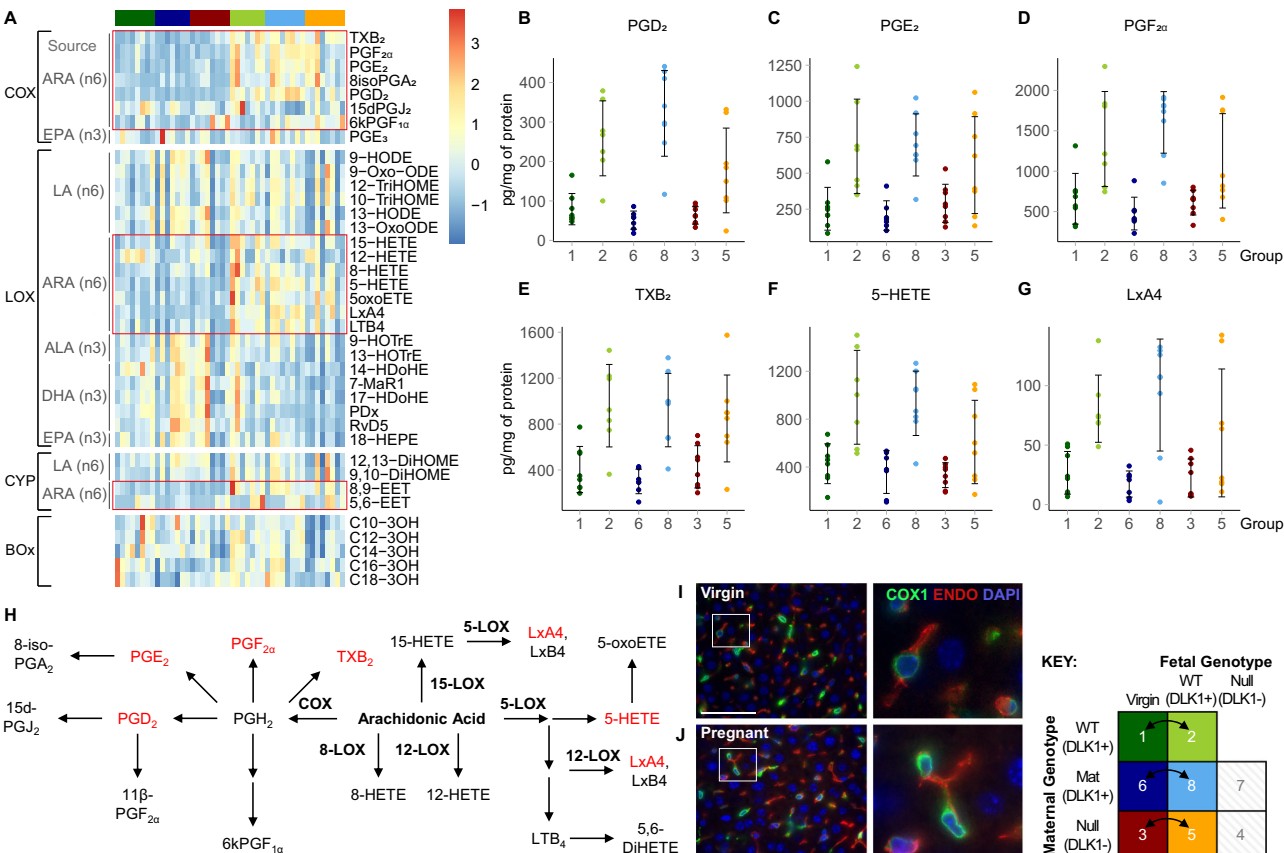

**Fig. 4 | Hepatic profile of PUFA metabolites in virgin and pregnant mice.**
**A** Heatmap showing all PUFA metabolite concentrations measured in the livers of virgin and pregnant (15.5 dpc) groups. Rows are organised by synthesis pathway and further sub-divided by the precursor PUFA from which the metabolite is sourced. Red boxes indicate ARA-derived metabolites. Values are z-transformed across samples. **B–G** PUFA metabolites identified as candidate biomarkers (CBMs) that distinguish pregnant from virgin livers. CBMs are classified as metabolites that passed both Bonferroni-adjusted two-tailed *t*-tests (*p*-value threshold = 0.00811) and sparse partial least squares discriminant analysis. CBM data is shown as individual concentration values with ± SD error bars. CBM tests were performed individually per genotype-matched virgin vs pregnant comparison. **H** Schematic tree diagram of all measured metabolites in the LOX and COX pathways that are derived from ARA. Metabolites identified as CBMs of pregnancy are indicated in red. *n* = 8

(group 1), *n* = 7 (group 2), *n* = 8 (group 3), *n* = 8 (group 5), *n* = 7 (group 6), *n* = 8 (group 8); mice per group. **I–J** Immunohistochemistry showing COX1 and endo-mucin co-expression in an independent cohort of virgin (**I**) and pregnant (**J**) livers. Scale bars represent 500 μm for low-magnification images and 50 μm for enlarged images. Immunohistochemistry experiments were repeated twice (*n* = 3 mice per condition). All metabolites are defined in the Materials and Methods. Mean concentrations of individual metabolites per group are found in Supplementary Data Table S8. ALA α-Linolenic acid, ARA arachidonic acid, BOx beta-oxidation, COX cyclooxygenase, CYP Cytochrome P450, DHA docosahexaenoic acid, ENDO endomucin, EPA eicosapentaenoic acid, LA linoleic acid, LxA4 lipoxin A4, LOX lipoxygenase, PGD₂ prostaglandin D₂, PGE₂ prostaglandin E₂, PGF₂α, prostaglandin F₂α, TxB₂ thromboxane B₂, 5-HETE 5-hydroxyeicosatetraenoic acid. Source data are provided as a Source Data file.

Data Table S9). Using a targeted approach, the dataset was assessed for transcriptional changes to rate-limiting steps within candidate pathways of interest, as depicted in Fig. 5D (Supplementary Data Table S10). Target genes were then validated by real-time quantitative PCR (RT-qPCR) analysis in virgin and pregnant liver samples from the current study (Fig. 5E–I).

We first investigated the source of hepatic LC-PUFAs in pregnancy. ARA and DHA may be directly imported into the liver from the diet or adipose tissue mobilisation, or they may be synthesised from n-3 and n-6 PUFA precursors using the LC-PUFA synthesis pathway[9]. For either case, the hepatic ARA and DHA pool requires FA influx, and we observed a two-fold increase in maternal gene expression of CD36, the major regulatory transporter of hepatic fatty acid uptake (Fig. 5D, F). Gene expression of fatty acid transfer proteins were unchanged (*Slc27a2*) or decreased (*Slc27a5*) in the microarray dataset (Fig. 5D). Crucially, *Fads1* and *Fads2* were upregulated in the maternal liver, which encode the rate-limiting delta-5 and delta-6 desaturases in LC-PUFA synthesis (Fig. 5D, F). LC-PUFA elongation enzymes were either unchanged (*Elovl2*), or down-regulated (*Elovl5*) (Fig. 5D, F). The rate-limiting desaturase in monounsaturated fatty acid formation,

*Scd1*, was considerably down-regulated in pregnancy, while no other changes were observed in the de novo fatty acid synthetic pathway (Fig. 5D, F), consistent with the lipidomics data that shows a selective increase in LC-PUFAs. These findings together support the influx and subsequent desaturation of n-3 and n-6 PUFA precursors in the maternal liver.

The hepatic pool of ARA and DHA are incorporated into lipids through synthetic pathways. The Kennedy Pathway describes the de novo synthesis of both triglycerides and phospholipids, wherein phosphatidic acid (PA) is generated as a common precursor[25]. The initial committing step in this pathway was upregulated in the pregnant liver (*Gpam;* Fig. 5G). For the second PA-generating step, the two genes for major hepatic PA-acyltransferases were oppositely regulated by pregnancy. *Agpat2*, encoding LPA-acyltransferase 2 was down-regulated, whereas *Agpat3* was induced (Fig. 5D, G). Importantly AGPAT2 and AGPAT3 have been reported to incorporate different fatty acids into PA, with AGPAT3 shown to selectively generate DHA-PA[26]. Diglycerides generated from PA are then used in the 3-step PC and PE synthetic pathways. No difference was observed for the initial committing steps (*Chka, Etnk1;* Fig. 5G), but the second rate-limiting step in

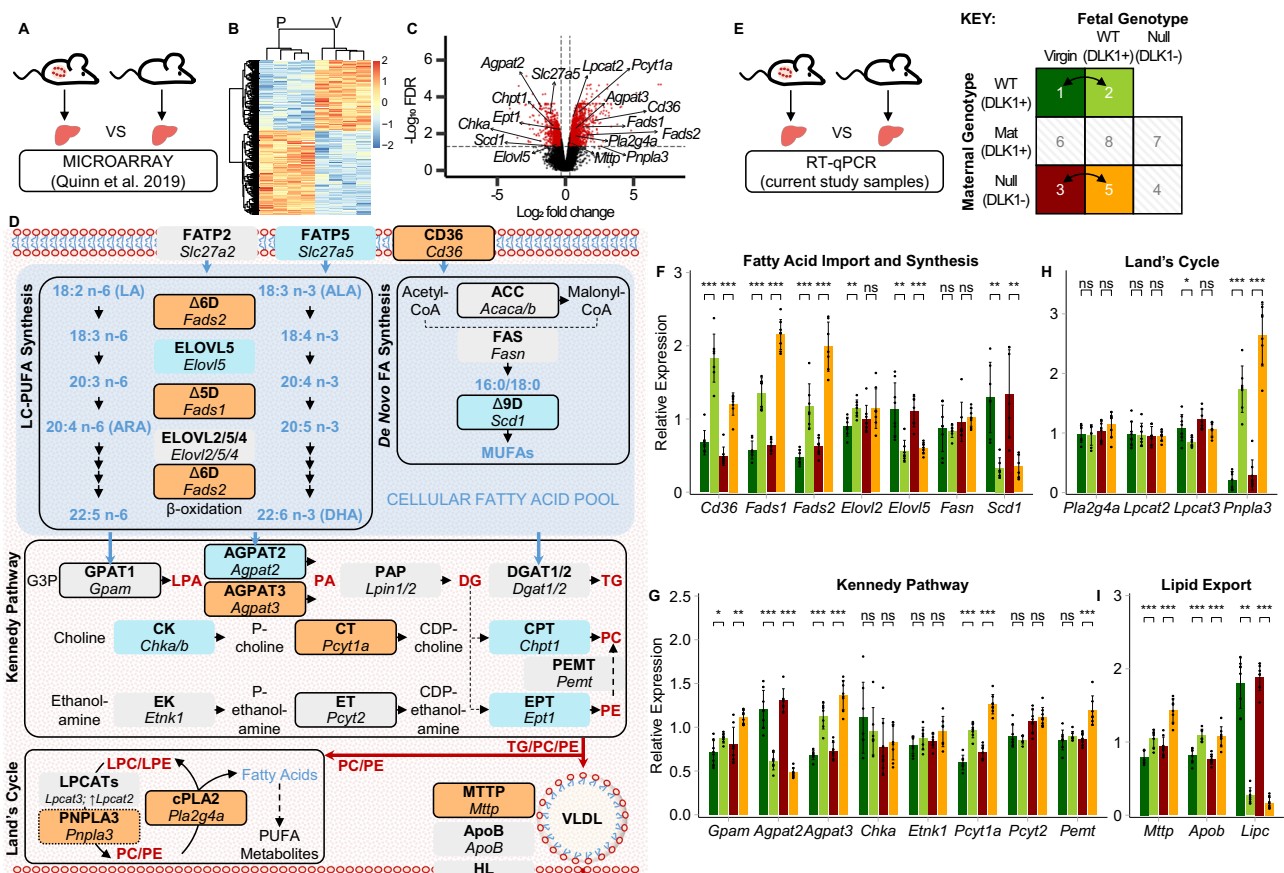

**Fig. 5 | Transcriptional analysis of LC-PUFA-phospholipid biosynthetic pathways in the pregnant liver. A** Schematic depicting the re-analysis of a published whole-transcriptome microarray dataset of livers from pregnant (14.5 dpc) and virgin mice (*n* = 4 mice per condition). **B** Heatmap showing z-transformed values of 3272 differentially expressed genes (DEGs) based on a p-value (corrected for multiple hypothesis testing based on the Benjamini–Hochberg procedure) threshold of 0.05 and a fold change threshold of 1.25. Differential expression analysis was performed using limma. List of DEGs is found in Supplementary Data Table S9. **C** Volcano plot showing DEGs (red) annotated with biosynthetic genes of interest. **D** Diagrammatic summary of hepatic lipid biosynthetic pathways chosen for targeted transcriptomics analysis annotated with microarray expression data (orange box = DEG that increased in pregnant livers; blue box = DEG that decreased in pregnant livers; grey box = gene not identified as differentially expressed). Known rate-limiting steps are indicated by thick black borders. Gene symbols are defined in Supplementary Data Table S10. **E** Schematic depicting real-time quantitative PCR (RT-qPCR) validation of candidate gene expression in livers from two genotype-matched virgin vs pregnant group comparisons. **F−I** RT-qPCR data is split into fatty acid import and synthetic pathway genes (**F**), Kennedy pathway genes (**G**), Land's

Cycle genes (**H**) and lipid export pathway genes (**I**). RT-qPCR data was normalised to housekeeping gene expression (*Tuba1*, *Tbp* and *Hprt*) and is shown as mean relative expression ± SD. Groups were called significantly different by two-tailed Mann-Whitney U tests (*$p$-value < 0.05; **$p$-value < 0.01; ***$p$-value < 0.001). $p$-values for *Cd36*: 1vs2 = 0.0003, 3vs5 = 0.0002; *Fads1*: 1vs2 = 0.0003, 3vs5 = 0.0002; *Fads2*: 1vs2 = 0.0006, 3vs5 = 0.0003; *Elovl2*: 1vs2 = 0.0041; *Elovl5*: 1vs2 = 0.0012, 3vs5 = 0.0006; *Scd1*: 1vs2 = 0.0012, 3vs5 = 0.0012; *Gpam*: 1vs2 = 0.0205 0.007; *Agpat2*: 1vs2 = 0.0003, 3vs5 = 0.0003; *Agpat3*: 1vs2 = 0.0003, 3vs5 = 0.0002; *Pcyt1a*: 1vs2 = 0.0006, 3vs5 = 0.0002; *Pemt*: 3vs5 = 0.0003; *Lpcat3*: 1vs2 = 0.0205; *Pnpla3*: 1vs2 = 0.0006, 3vs5 = 0.0002; *Mttp*: 1vs2 = 0.0035, 3vs5 = 0.0003; *Apob*: 1vs2 = 0.0003, 3vs5 = 0.0003; *Lipc*: 1vs2 = 0.0003, 3vs5 = 0.0003. *n* = 8 (group 1; exceptions: *n* = 7 for *Scd1*, *Elovl2*, *Fads2*, *Chka*, *Pcyt1a*, *Mttp*, *Pnpla3*), *n* = 7 (group 2; exceptions: *n* = 6 for *Scd1*), *n* = 8 (group 3; exceptions: *n* = 7 for *Scd1*, *Fasn*, *Lipc*, *Elovl5*, *Agpat2*, *Lpcat2*, *Lpcat3*), *n* = 8 (group 5; exceptions: *n* = 7 for *Elovl5*, *Fads2*, *Etnk1*, *Pemt*, *Lpcat2*; *n* = 6 for *Scd1*); mice per group. DG diglyceride, LPA, *lyso*-phosphatidic acid, LPC *lyso*-phosphatidylcholine, LPE *lyso*-phosphatidylethanolamine, PA phosphatidic acid, PC phosphatidylcholine, PE phosphatidylethanolamine, TG triglyceride. Source data are provided as a Source Data file.

PC synthesis was increased in the pregnant liver (*Pcyt1a;* Fig. 5D, G). The equivalent rate-limiting step in PE synthesis was unchanged (*Pcyt2;* Fig. 5D, G). Thus, selective adaptations to the Kennedy Pathway may promote AGPAT3-mediated synthesis of DHA-containing PCs in pregnancy.

While incorporation of DHA into phospholipids is thought to occur mainly as a result of the Kennedy pathway, ARA-containing phospholipids are predominantly generated by the Land's cycle, which cleaves and re-acylates fatty acids on the *sn*-2 position of fully synthesised phospholipids[25]. Incorporation of ARA into phospholipids may be achieved by activation of re-acylation enzymes, of which *Lpcat3* has known specificity for ARA[27,28]. However, we observed only small shifts in the expression of genes encoding known phospholipid acyltransferase enzymes (Fig. 5D, H). Enzymes that cleave the *sn*–2-

position of phospholipids, phospholipase A₂ (PLA₂) lipases, are members of a large, complex and not fully characterised protein family[29,30]. Research into PLA₂ function has largely focused on their role in liberating ARA for PUFA metabolite synthesis, for which the PLA₂ isoform with specificity to ARA in the *sn*−2 position, encoded by *Pla2g4a*, catalyses the known rate limiting step. Elevated expression of *Pla2g4a* in the pregnant liver (Fig. 5D) is consistent with the observed pregnancy-specific rise in ARA-derived metabolites (Fig. 4A). PLA₂ isoforms involved in cleaving other *sn*−2 fatty acids from phospholipids for subsequent ARA incorporation are not established. Two other PLA₂-encoding genes, *Pla2g7* and *Pla2g15* were upregulated in the pregnant liver (Supplementary Data Table S10), yet their substrate specificities and roles have not been fully characterised. Therefore, we were unable to uncover any evidence for selective increase in the Land's cycle

activity that favour increased ARA incorporation in phospholipid of the pregnant liver, at the transcriptional level.

We additionally considered alternative phospholipid synthetic pathways, including the liver-specific conversion of PE into PC via phosphatidylethanolamine methyl transferase (PEMT). Both gene and protein expression were unchanged in pregnant livers (Fig. 5D, G; Supplementary Fig. S4A, B). Of note, *Pnpla3* was ~5-fold upregulated in the pregnant liver (Fig. 5D, H). This gene encodes a fatty acid remodelling enzyme which has recently been shown to transfer LC-PUFAs (including DHA and ARA) from liver triglycerides to the phospholipid compartment via alternative re-acylation pathways[31].

Ultimately, the parallel rise in triglycerides and LC-PUFA-phospholipids, particularly PC, between liver and plasma supports the lipoprotein-mediated export of these lipids. Gene expression of VLDL-specific proteins, ApoB and MTTP, were increased in the maternal liver, while that of hepatic lipase (*Lipc*), which hydrolyses circulating VLDL, was decreased (Fig. 5D, I). Taken together, our targeted transcriptional analysis demonstrates the regulated adaptations to maternal hepatic lipid pathways that promote PUFA influx, PUFA-specific desaturation, de novo biosynthesis of LC-PUFA-containing PCs and lipid export pathways.

### LC-PUFA-phospholipid synthesis is transcriptionally co-regulated by the liver X receptor in late pregnancy

In our RT-qPCR analysis that compared two genotype-matched virgin vs pregnant livers, one pregnant group lacked DLK1 derived from maternal tissues (group 5; Fig. 5E-I). We noticed that for many LC-PUFA-phospholipid biosynthetic genes that were altered by pregnancy, the effect was greater in this group compared to the wild-type pregnant mice (group 2). Considering the observed effect of DLK1 on PC and PE class levels (Fig. 2I, J), and on the ARA-PE species (PE(18:0/20:4); Fig. 3G), we investigated whether DLK1 affects the transcriptional adaptations promoting LC-PUFA-phospholipid synthesis in the late gestational liver.

Pregnant mice lacking maternal-derived DLK1 exhibited increased gene expression of both rate-limiting PUFA desaturase enzymes (*Fads1, Fads2;* Fig. 6A), the rate-limiting step in PC biosynthesis (*Pcyt1a;* Fig. 6B), the TG-phospholipid trans-acylase (*Pnpla3*, Fig. 6C) and the rate-limiting MTTP protein required for lipoprotein assembly (*Mttp;* Fig. 6D). Alongside trends in other Kennedy pathway genes including increased *Gpam* and reduced *Agpat2* expression, this DLK1 effect notably augments the transcriptional response of normal pregnancy. We further investigated this coordinated gene response by running a multiple correlation analysis of these selected genes in pregnant and virgin mice (Fig. 6E, F). The significant and similar correlations between LC-PUFA-phospholipid biosynthetic genes in pregnant mice, compared to that of virgins, suggests this pathway is a transcriptionally co-regulated process in pregnancy that can be modulated by the maternal DLK1 genotype.

To explore potential upstream regulators involved in the coordinated LC-PUFA-phospholipid transcriptional response, we assessed the differentially expressed genes derived from the virgin and pregnant hepatic microarray re-analysis for enrichment of transcription factor targets using ChIP enrichment analysis (ChEA)[32]. The upregulated gene list from the maternal hepatic transcriptome was significantly enriched for 10 transcription factor pathways (Table 3, Supplementary Data Table S11), of which only the Liver X receptor (LXR), retinoid X receptor (RXR) and Estrogen Receptor 1 (ESR1) pathways induced genes involved in the coordinated maternal LC-PUFA-phospholipid response. Importantly, 6 key LC-PUFA-phospholipid genes (*Cd36, Fads1, Fads2, Pcyt1a, Pnpla3, Mttp*) were identified as targets of the master lipid regulator, LXR. Increased gene expression of the inducible LXRa isoform (*Nr1h3*) was confirmed in pregnant livers in both the re-analysed microarray dataset and in current study samples (Supplementary Data Table S10; Supplementary Fig. 5A). However, classical hepatic LXR

targets involved in reverse cholesterol transport (*Cyp7a1, Abcg5, Abcg8, Abca1, Pltp, Lpl*), and de novo lipogenesis (*Srebf1, Fasn, Scd1*) were unchanged or downregulated in pregnancy (Supplementary Data Table S10; Supplementary Fig. 5B, C).

Reduced gene expression of classical lipogenic and cholesterol LXR targets in late pregnancy was previously described by Nikolova and colleagues[33], where both wild type (WT) and *Lxrab*[-/-] double knockout (DKO) mutant dams showed this late-pregnancy associated reduction in LXR target transcription. Using liver samples from the Nikolova cohort[33], we quantified the expression of the six putative LXR-induced LC-PUFA-phospholipid genes at early and late gestational timepoints (Fig. 7B–G). LXR DKO mice demonstrated a blunted late-gestational rise in *Fads1, Fads2, Pnpla3* and *Cd36* expression, with a non-significant trend also evident for *Pcyt1a*. Measurement of other key LC-PUFA-phospholipid genes revealed a similar attenuated late-pregnancy increase in *Agpat3* expression (Fig. 7H) but no change to the fall in *Agpat2* expression (Supplementary Fig. S5D). Reduced expression of *Mttp, Lpcat3* and *Gpam* in LXR DKO mice at all timepoints suggests them to be pregnancy-independent LXR targets (Fig. 7G; Supplementary Fig. S5E, F).

To test whether LXR-manipulation ultimately influences LC-PUFA-phospholipid levels in pregnancy, we performed targeted LC-MS/MS in samples from the Nikolova cohort at late-gestational timepoints, specifically measuring the concentrations of the ARA and DHA-containing PC and PE lipids (Supplementary Data Table S12). At 14.5 dpc, LXR DKO mice have a lower concentration of DHA-PC lipids in serum alongside a non-significant fall in the liver (Fig. 7I, J). A non-significant trend was also evident for ARA-phospholipids in serum. No shift was observed for these lipids at the pre-parturition 18.5 dpc timepoint, which in agreement with the observed maximal expression of LC-PUFA-phospholipid synthetic genes at 14.5 dpc compared to 18.5 dpc (Fig. 7B–H), suggests that the LC-PUFA-phospholipid pathway is most active at the earlier peak catabolic phase of gestation.

Finally, we hypothesised that LXR-activation of the LC-PUFA-phospholipid transcriptional program in the maternal liver is a critical adaptive pathway that facilitates LC-PUFA delivery to the fetus (Fig. 7A). We tested this hypothesis by performing targeted LC-MS/MS in fetal tissues from WT and LXR DKO dams at 18.5 dpc. While no significant changes were observed in placentas from DKO dams compared to those from WT mice (Fig. 7K), DHA-PC concentrations were lower in fetal livers from DKO dams (Fig. 7L). At the species level, the most abundant DHA-containing PC lipid, PC(38:6), which accounts for over 50% of all DHA-PC, was reduced in both maternal liver and serum at 14.5 dpc, and in the 18.5 dpc fetal liver (Supplementary Data Table S12). We also observed lower concentrations of ARA-PE in the fetal liver from DKO dams, with a non-significant fall also evident for ARA-PC. DHA-PE lipid concentrations remained unchanged across all tissues in LXR DKO mice. Our study ultimately defines an LXR-mediated pathway that selectively enriches LC-PUFAs into phospholipids to transfer ARA and DHA to the fetal compartment in late pregnancy.

## Discussion

Our findings challenge the long-established role of triglycerides as the major lipid fraction that supplies LC-PUFAs to the fetus[7,34]. Recent work has highlighted that phospholipids as well as triglycerides may contribute to the maternal supply of fetal LC-PUFAs; for example, administration of isotope-labelled fatty acids to pregnant women prior to cesarean section revealed that PUFAs (LA and DHA) were found in both the triglyceride and phospholipid fractions[35]. Studies on rat dams in late gestation show that hepatic and serum phospholipids contain a higher proportion of LC-PUFAs compared to triglycerides[20,21]. These findings together build a more complex picture of the maternal lipidome, where raised triglycerides

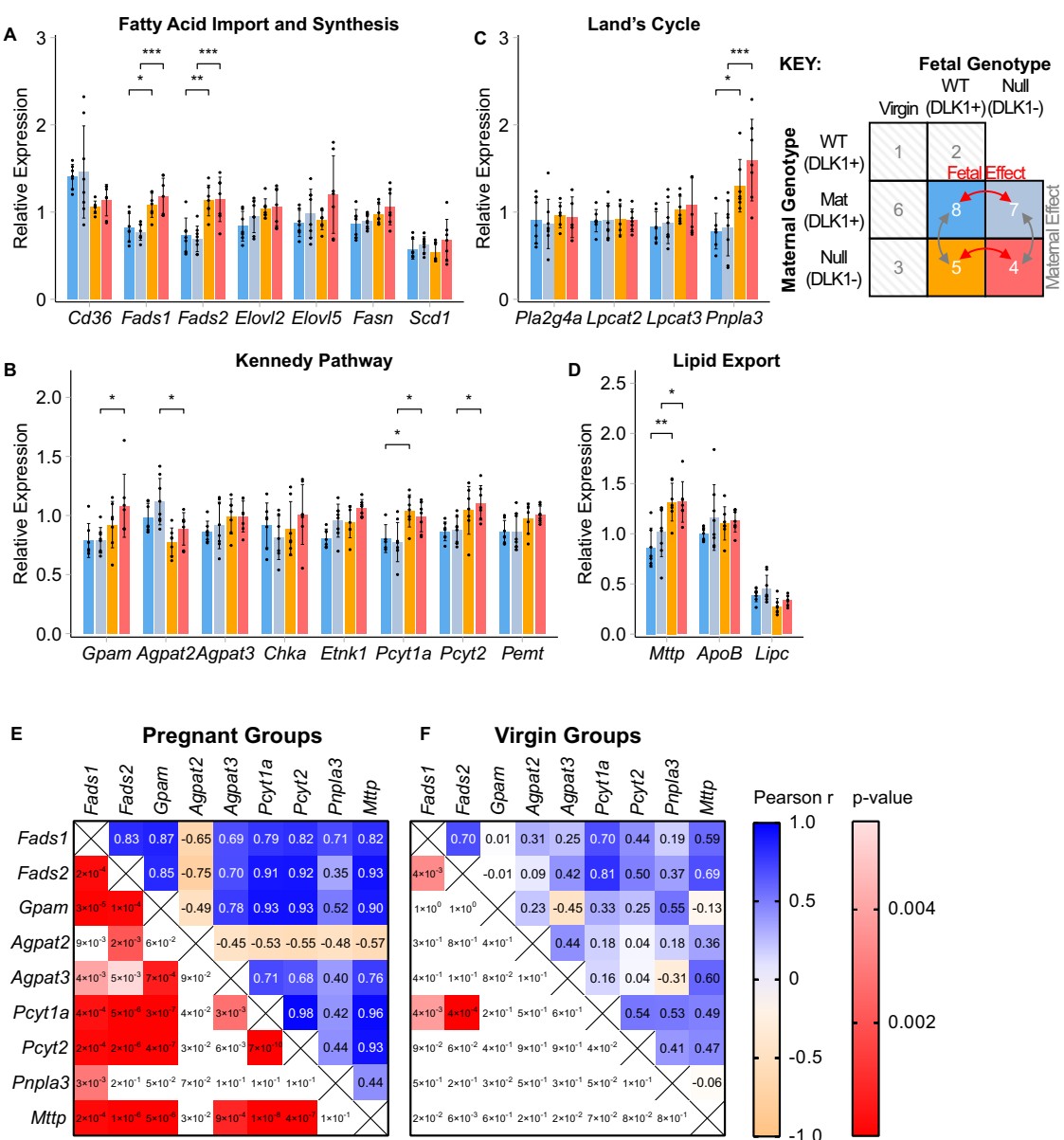

**Fig. 6 | Coordinated hepatic expression of LC-PUFA-phospholipid biosynthetic genes in response to *Dlk1*-manipulation in pregnancy. A–D** Real-time quantitative PCR (RT-qPCR) analysis of LC-PUFA-phospholipid biosynthetic genes in pregnant mice (15.5 dpc) in response to *Dlk1* manipulation. Genotype-matched replicate comparisons were performed to assess the effect of maternal-derived DLK1 protein or fetal-derived DLK1 protein on the expression of fatty acid import and synthetic pathway genes (**A**), Kennedy pathway genes (**B**), Land's Cycle genes (**C**) and lipid export pathway genes (**D**). Gene symbols are defined in Supplementary Data Table S10. RT-qPCR data was normalised to housekeeping gene expression (*Tuba1* and *Hprt*) and is shown as mean relative expression ± SD. Gene expression was compared between all four experimental groups by one-way ANOVA with Tukey's multiple-comparison post-hoc test (*$p$-value < 0.05; **$p$-value < 0.01; *** $p$-value < 0.001). $p$-values for *Fads1*: 7vs4 = 0.0002, 8vs5 = 0.0226; *Fads2*: 7vs4 = 0.0008,

8vs5 = 0.0041; *Gpam*: 7vs4 = 0.0262; *Agpat2*: 7vs4 = 0.0292; *Pcyt1a*: 7vs4 = 0.0348, 8vs5 = 0.0423; *Pcyt2*: 7vs4 = 0.019; *Pnpla3*: 7vs4 = 0.0008, 8vs5 = 0.0363, *Mttp*: 7vs4 = 0.0474, 8vs5 = 0.0023. $n$ = 7 (group 4; exceptions: $n$ = 5 for *Pla2g4a*), $n$ = 7 (group 5; exceptions: $n$ = 6 for *Pla2g4a*, $n$ = 5 for *Lpcat2*), $n$ = 8 (group 7; exceptions: $n$ = 7 for *Elovl2*), $n$ = 7 (group 8; exceptions: $n$ = 5 for *Pcyt1a*); mice per group. **E, F** Two-tailed multiple Pearson's correlations between DLK1-responsive genes in pregnant (groups 2 and 5; **E**) and virgin (groups 1 and 3; **F**) livers. $p$-values in shaded red boxes indicate statistically significant correlations (Bonferroni-adjusted $p$-value threshold = 0.0045). $n$ = 8 (group 1; exceptions: $n$ = 7 for *Fads2*, *Pcyt1a*, *Pnpla3*), $n$ = 7 (group 2), $n$ = 8 (group 3; exceptions: $n$ = 7 for *Agpat2*), $n$ = 8 (group 5; exceptions: $n$ = 7 for *Fads2*); mice per group. Source data are provided as a Source Data file.

increase the availability of all fatty acids for the fetus, while ARA and DHA are selectively channeled into phospholipids to meet the profound LC-PUFA requirements of late pregnancy. This is important because it suggests that the control point of biomagnification is in the regulation of LC-PUFA phospholipid supply by the mother.

Despite the long-standing recommendation to consume DHA-rich foods during pregnancy[36], the combined supplementation evidence does not support long-term cognitive or visual benefits in the child[37–41]. By contrast, positive associations have been observed between DHA levels in maternal plasma and numerous postnatal outcomes[9]. Although endogenous LC-PUFA synthesis is long-thought to only minorly contribute to total mammalian ARA and DHA requirements[12], the findings by us and others support an adaptive importance for the endogenous synthetic pathway in the pregnant liver, in order to meet the LC-PUFA needs of the fetus.

**Table 3 | Transcription factor pathways enriched for genes that are upregulated in the pregnant liver**

| Transcription Factor Pathway | Adjusted P-Value | #Upregulated DEGs (/#All Targets) | LC-PUFA-Phospholipid Pathway DEGs | |
|---|---|---|---|---|
| | | | # | Gene Names |
| FOXM1 | 1.05E-11 | 56/267 | 0 | |
| IRF8 | 1.04E-10 | 217/2000 | 0 | |
| SMRT | 2.45E-05 | 193/2000 | 0 | |
| RXR | 7.03E-05 | 190/2000 | 3 | Fads1, Fads2, Pnpla3 |
| LXR | 1.33E-04 | 188/2000 | 6 | Cd36, Fads1, Fads2, Pcyt1a Pnpla3, Mttp |
| MECOM | 2.76E-04 | 182/1951 | 0 | |
| NCOR | 4.94E-04 | 184/2000 | 0 | |
| CLOCK | 1.71E-02 | 47/407 | 0 | |
| ESR1 | 1.78E-02 | 50/444 | 1 | Fads2 |
| IRF8 | 2.91E-02 | 38/319 | 0 | |

Overlap analysis between transcription factor targets and upregulated differentially expressed genes (DEGs) generated from the re-analysis of a published whole-transcriptome microarray dataset of livers from pregnant (14.5 dpc) and virgin mice (n = 4 mice per condition), using ChIP enrichment analysis (ChEA) through the Enrichr platform. Enriched DEGs were then searched for LC-PUFA-containing-phospholipid biosynthetic genes of interest. See Supplementary Data Table S11 for the full list of enriched DEGs in each transcription factor pathway. Differential expression analysis was performed using limma with a p-value (corrected for multiple hypothesis testing based on the Benjamini–Hochberg procedure) threshold of 0.05 and a fold change threshold of 1.25 (DEG list is found in Supplementary Data Table S9).

We propose two key synthetic pathways that most likely underlie the raised availability of LC-PUFA-phospholipids in pregnancy. First, we report an upregulation of LC-PUFA synthesis mediated by the delta-5 and delta-6 desaturases (encoded by *Fads1* and *Fads2*), which has previously been reported in rat pregnancy[20,42,43]. The critical role of both desaturases at the transcriptional level is emphasized by the numerous associations between maternal polymorphisms within the *FADS1* and *FADS2* genes with PUFA concentrations in maternal and fetal blood[44–47]. Second, we propose a pregnancy-associated selective modification to the Kennedy Pathway. In addition to the increased expression of the initial committing step that generates LPA (*Gpam*), we report an isoform switch from *Agpat2* to *Agpat3* expression, which encode acyltransferases that incorporate the second fatty acid onto LPA to generate PA, the precursor molecule to triglycerides and phospholipids. AGPAT2 is a highly expressed liver enzyme with well-established function in the Kennedy Pathway. Its preferential incorporation of LA (18:2 n-6)[26] may underlie the fall in LA-containing phospholipids observed in the current study. Although less well characterised, AGPAT3 displays distinct selectivity towards DHA[48,49]. Recently Hishikawa and colleagues reported reduced levels of DHA in liver phospholipids in mice with a liver-specific deletion of *Agpat3*[50]. We therefore propose an adaptive mechanism in the maternal liver that targets the Kennedy Pathway to promote the synthesis of DHA-phospholipids in pregnancy.

The concurrent changes to the same PC species between liver and plasma support a particular role for PC, which has long been reported as an LC-PUFA storage lipid[51,52], as the major transport vehicle of LC-PUFAs. Further evidence of this role can be inferred from choline investigations, which like DHA, is critical for neurodevelopment and enriched in fetal cord blood[53]. Isotope-tracing in pregnant women has shown that choline enrichment in the fetal compartment is principally derived from PC that was synthesized in the maternal liver[54]. Notably, this study also reported that the majority of this PC was synthesized via the PEMT pathway, building on the hypothesis that PEMT, an estrogen-responsive enzyme[55] that selectively synthesizes PCs that contain LC-

PUFAs[56], is the major route for choline and DHA delivery to the fetus[57]. In contradiction to this theory, we report an upregulation to the rate-limiting PC synthesis step of the Kennedy pathway (*Pcyt1a*), while no changes were observed to PEMT at the gene or protein level. Therefore, we suggest that the maternal transfer of choline and DHA is driven at least in part by the Kennedy pathway, which targets *Agpat3* and *Pcyt1a* to selectively generate DHA-PC.

In pregnancy hepatic ARA was elevated in PC but reduced in PE compared to virgin livers, implicating enhanced *sn*−2 remodeling (Land's Cycle) in the maternal liver. The best characterized acyl-transferase in this pathway, LPCAT3, shows substrate specificity towards linoleic acid and ARA[28,58,59], but a transcriptional shift in *Lpcat3* was not observed. However, we report increased expression of PNPLA3[60]. A recent knockout study proposed that PNPLA3 transfers LC-PUFAs from triglycerides to phospholipids[31], supporting the emerging significance of alternative remodeling pathways to the Land's Cycle in LC-PUFA-phospholipid synthesis[25]. Ultimately, the generation of ARA-phospholipids by LPCAT3[27,28] and potentially by PNPLA3[61] is vital for the normal production of hepatic VLDLs. Given the requirement of PC for normal VLDL secretion[62], the maternal liver may selectively synthesize ARA-PC to promote VLDL-mediated lipid export into the circulation.

The observed increase in the hepatic synthesis and export of VLDLs, which consist of a triglyceride core surrounded by a PC-rich phospholipid monolayer[63], supports lipoprotein-mediated transport of hepatic triglycerides and LC-PUFA-PCs in late pregnancy. The placental uptake of PUFAs from lipoproteins is further supported by the placental expression of lipoprotein lipase (LPL) and endothelial lipase (EL). EL is the major lipase found on the maternal blood-facing surface of trophoblast cells in the human placenta[64]. In contrast, LPL, the major TG lipase, is expressed at lower levels than EL in villous cytotrophoblasts and villous syncytiotrophoblast cells in early pregnancy, and not present at term[64]. EL and LPL are both expressed in the exchange compartment of the mouse placenta[65]. EL is primarily a phospholipase A1[66], thus its activity will release *sn*−1 (primarily saturated) FAs with a *lyso*-phospholipid remnant containing a *sn*−2 (commonly unsaturated) FA. Hence, to prioritize both LC-PUFA and choline transfer, the placenta may transport the LC-PUFA-containing *lyso*-phospholipids using major facilitator super-family domain 2 protein (MFSD2A), a newly-described *lyso*-PC symporter with high affinity for LPC(22:6)[67]. Genetic deletion investigations in mice have recognized MFSD2A as a critical means of DHA supply to the brain and retina[67,68], and clinical correlations were recently reported between placental MFSD2A expression and DHA levels in cord blood[69,70]. Interestingly, we observed a 2-fold reduction in *Mfsd2a* expression in pregnant livers which may suggest a complementary reduction in maternal hepatic DHA-LPC uptake (Supplementary Data Table S10, Supplementary Fig. S4C).

By constructing a PUFA metabolite panel of the pregnant mouse liver, we identified raised concentrations of metabolites that were specifically derived from ARA, suggesting that the targeted de-acylation of ARA is also a crucial source for the maternal PUFA metabolite pathway. Broadly divided into the ARA-derived inflammatory mediators and the n3-PUFA-derived anti-inflammatory metabolites[71], our findings further describe the characteristic inflammatory phenotype of pregnancy. We observed a particular enrichment of classical prostanoids in the COX pathway, which have diverse housekeeping and inflammatory actions. The expression of the COX-1 isoform in hepatic endothelial cells particularly points towards their hemodynamic properties in the maternal liver[72]. Given the reported associations between prostanoids and pre-eclampsia[73–75], and their critical regenerative roles following hepatic injury[76], further research into maternal hepatic bioactive PUFA metabolites may elucidate underlying mechanisms that implicate liver dysfunctions to pregnancy complications.

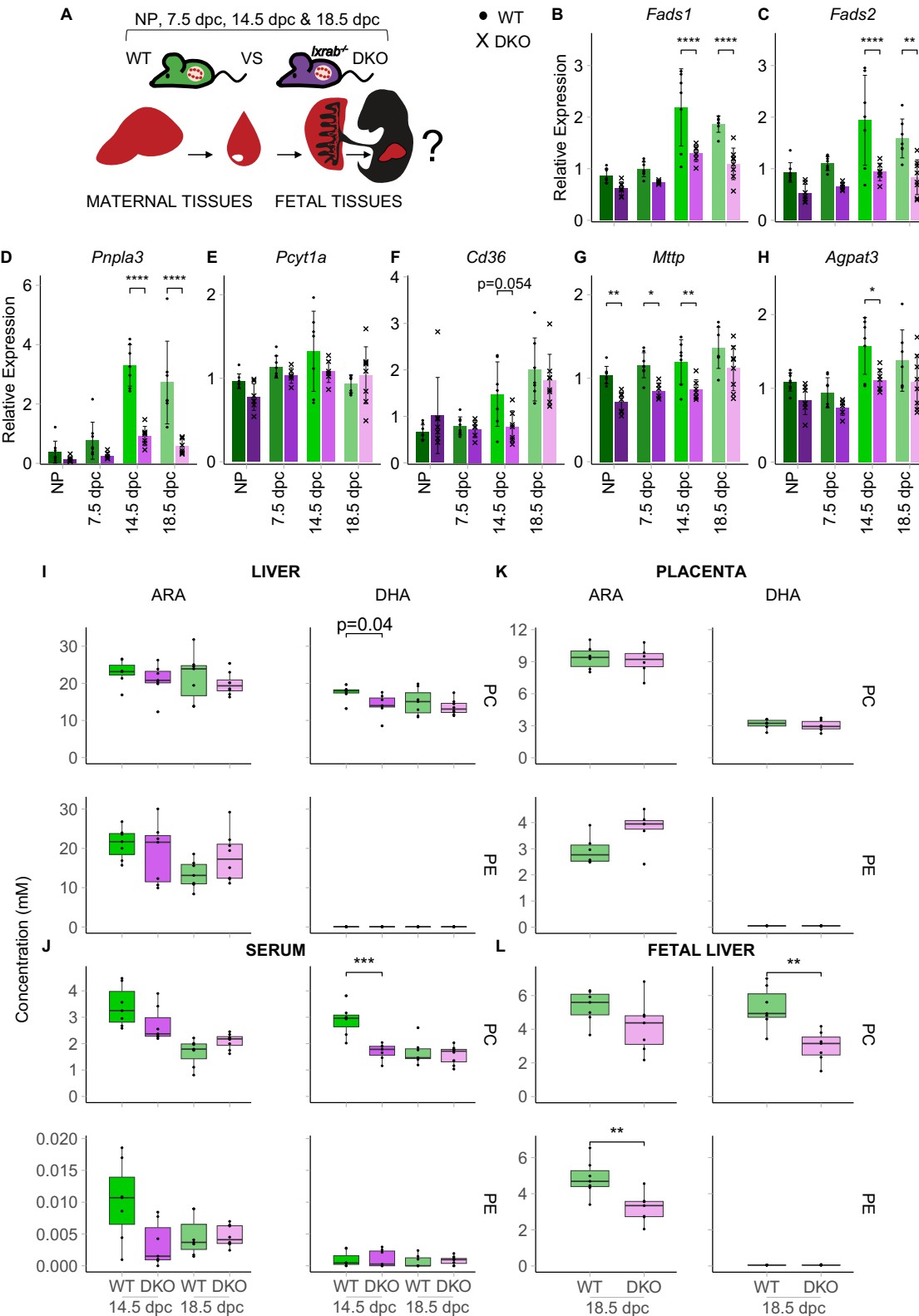

In our previous study we found that fetally-derived DLK1 in the maternal circulation had a minimal impact on fed metabolism[4]. Consistently, here fetal DLK1 led to a small shift in total PE:PC ratios (Fig. 2I, J). However, the ablation of *Dlk1* in maternal tissues modified the dynamics of lipid mobilisation, potentially via indirect effects on whole body adipose storage. We found that dams lacking DLK1 derived from maternal tissues had modestly increased transcription of rate-limiting steps in the hepatic synthesis and export of LC-PUFA-

phospholipids. Notably, in our previous study, these mice commenced pregnancy with a higher adipose mass, but exhibited a reduced ability to gain adipose tissue during pregnancy[4], and appear to accumulate less hepatic TG (Fig. 2B). Since dietary n-3 and n-6 PUFAs are preferentially stored in adipose tissue[77], this compensatory upregulation in the liver suggests maternal adipose tissue remodelling is a critical source of the longer-chained PUFAs for hepatic ARA and DHA synthesis.

**Fig. 7 | LXR-mediated activation of hepatic LC-PUFA-phospholipid biosynthesis, export and fetal transfer in late pregnancy. A** Schematic of the tissues analysed from an independent cohort of wild-type (WT) and $Lxrab^{-/-}$ (LXR double knockout (DKO)) mice at non-pregnant (NP) and various gestational timepoints (7.5, 14.5 and 18.5 dpc). Arrows indicate the hypothesized transfer of LC-PUFA-containing phospholipids between maternal and fetal compartments. **B**–**H** RT-qPCR analysis of hepatic LC-PUFA-phospholipid biosynthetic genes in WT and LXR DKO groups. RT-qPCR data was normalised to housekeeping gene expression (*Tuba1*, *Tbp* and *Hprt*) and is shown as mean relative expression ± SD and were compared by two-way ANOVA with Šídák's multiple comparison (\*p-value < 0.05; \*\*p-value < 0.01; \*\*\*p-value < 0.001; \*\*\*\*p-value < 0.0001). WT vs DKO p-values for *Fads1*: 14.5 dpc = $6.28 \times 10^{-06}$, 18.5 dpc = $4.83 \times 10^{-05}$; *Fads2*: 14.5 dpc = $2.80 \times 10^{-05}$, 18.5 dpc = 0.0012; *Pnpla3*: 14.5 dpc = $1.50 \times 10^{-08}$, 18.5 dpc = $9.54 \times 10^{-08}$; *Mttp*: NP = 0.0056, 7.5 dpc = 0.0143, 14.5 dpc = 0.0066; *Agpat3*: 14.5 dpc = 0.0102. $n = 8$ (NP WT), $n = 7$ (NP DKO), $n = 8$ (7.5 dpc WT), $n = 6$ (7.5 dpc DKO), $n = 7$ (14.5 dpc WT), $n = 7$ (14.5 dpc DKO) $n = 7$ (18.5 dpc WT), $n = 8$ (18.5 dpc DKO); mice per condition.

**I**–**L** Grouped concentrations of ARA-containing PC, DHA-containing PC, ARA-containing PE and DHA-containing PE lipid species selected for targeted LC-MS/MS analysis in liver (**I**), serum (**J**), placenta (**K**) and fetal livers (**L**) from WT and LXR DKO groups at late-gestational timepoints. Individual lipid concentrations are found in Supplementary Data Table S12. Grouped lipid concentrations are presented as boxplots (median and IQR (25th and 75th percentiles) with whiskers showing 1.5\*IQR) with individual values and Bonferroni-adjusted t-tests (p-value threshold = 0.025) were performed for each WT vs DKO comparison (\*p-value < 0.05; \*\*p-value < 0.01; \*\*\*p-value < 0.001; \*\*\*\*p-value < 0.0001). p-values for (**J**) - DHA-PC: 14.5 dpc = 0.0009; (**L**) - DHA-PC: 18.5 dpc = 0.0021; (**L**) - ARA-PE: 18.5 dpc = 0.0060. $n = 7$ (14.5 dpc WT), $n = 7$ (14.5 dpc DKO), $n = 7$ (18.5 dpc WT; exceptions: $n = 6$ for placenta), $n = 8$ (18.5 dpc DKO; exceptions: $n = 6$ for placenta; $n = 7$ for fetal liver); mice per condition. ARA arachidonic acid, DHA docosahexaenoic acid, PC phosphatidylcholine, PE phosphatidylethanolamine. Source data are provided as a Source Data file.

The synchronized, yet selective, transcriptional response observed in our *Dlk1* model highlights the hepatic synthesis and export of LC-PUFA-phospholipids as a regulated maternal adaptation. Using several independent models of pregnancy, our study suggests this coordinated response to be mediated by LXR, an oxysterol-activated nuclear transcription factor that forms a heterodimer with retinoid X receptor (RXR) and acts as a master regulator of lipid metabolism[78]. Building on the observation from Nikolova and colleagues that classical LXR-stimulated lipid pathways, reverse cholesterol transport (RCT) and de novo lipogenesis, are suppressed in late gestation[33], our data proposes the hepatic LC-PUFA-phospholipid transcriptional program as an alternative LXR-induced pathway that is selectively activated in late pregnancy. The significance of this adaptive pathway is strengthened by its influence on LC-PUFA concentrations observed in the fetal liver. Amniotic-injection of PUFAs in late-gestational embryonic rats has been shown to rapidly accumulate in fetal liver and brain tissue, mainly within the phospholipid compartment[79], implicating the fetal liver as a major sink for circulating PUFAs.

We were initially surprised to find that maternal LXR-DKO resulted in reduction in LC-PUFA accumulation in the fetal liver at 18.5 dpc without apparent alteration to the levels of these FAs in the placenta. However, the lack of concordance between the placental and fetal compartment may be explained in several ways: Firstly, we see a higher overall concentration and a bigger differential between wild-type and LXR-DKO levels of plasma PL-DHA and ARA at 14.5 dpc compared to 18.5 dpc (Fig. 7J). This is consistent with the greater induction of expression of LXR target genes at the 14.5 dpc time-point (Fig. 7B-H). We measured lipids in the fetus and placenta at the end of gestation (18.5 dpc), representing the sum of total PL-DHA/ARA accumulation in the fetus over the course of pregnancy, but not the steady state level in the placenta. Secondly, recent studies employing mathematical models in combination with stable isotope tracing of FAs in a human placental perfusion system have demonstrated that the placenta accumulates a relatively low level of DHA compared to other FAs (e.g, roughly 10-fold less compared to 16:0,[80]. This is in agreement with historical findings that DHA is rapidly transported across the placenta with little placental accumulation. Interestingly an opposite effect is observed for n-6 PUFAs, particularly ARA which does accumulate in the placenta, possibly in reserve for prostanoid and endocannabinoid synthesis[81,82]. In keeping with these findings, in our data PC-associated DHA in the placenta reached only ~50% of the concentration in the fetal liver, replicating biomagnification (Fig. 7K). Moreover, placental PC-ARA concentrations exceed fetal liver levels, suggesting placental accumulation. How specific lipids and their FAs are partitioned to the placental metabolic pool or the fetal circulation is an extremely important topic, since these processes are modified by maternal diabetes and obesity[80,83]. There is also an increasing appreciation that

fetal/placental sex interacts with maternal diet to modulate placental adaptation to compromised maternal metabolic health[84,85]. Future studies where placental, fetal and maternal lipid metabolism are manipulated in isolation will be required to fully understand these important mechanisms.

Ultimately, our study demonstrates that LXR-induced phospholipid synthesis in the maternal liver represents a critical source of LC-PUFAs for the developing fetus.

This study used Direct-Infusion Mass Spectrometry (DI-MS) to generate an unbiased lipidomics profile of the pregnant mouse liver. Since this method is unable to distinguish the fatty acid moieties associated within individual lipids, assumptions were made based on their total carbon and double-bond content. We addressed this limitation using our candidate biomarker discovery pipeline and subsequent targeted profiling by liquid chromatography-mass spectrometry (LC-MS/MS), which allowed us to acquire the fatty acid compositions of hypothesis-based lipids of interest, with the additional benefit of DI-MS signal verification.

Recent work has highlighted an important role for fetal sex in placental metabolism and transfer of DHA in the context of maternal obesity. Specifically, male offspring of mothers with obesity have reduced placental and plasma levels of DHA compared to females, or the offspring of lean mothers[86,87]. Our study focuses on the maternal hepatic mechanism of biomagnification rather than the fetal/placental uptake route. Because rodent pregnancies contain multiple pups of both sexes, it is not possible to dissect the impact of fetal sex on the maternal hepatic response. Future work must address both placental compensatory mechanisms of reduced maternal LC-PUFA supply and should include offspring sex in this analysis.

The conclusions in our study are further limited by the incomplete comparability between mouse and human lipoprotein metabolism, which have been underscored by many decades of work by investigators attempting to model cardiovascular disease in rodent models. Mice do not spontaneously generate atherosclerotic plaques, due at least in part by their low VLDL/LDL cholesterol levels compared to HDL-C, and lack of the enzyme CETP. However, in contrast to knowledge about cholesterol metabolism, much less is appreciated about mechanisms and species differences in lipid transport in the lipoprotein compartment. Although they are known to be 'HDL dominant', mice carry a significant proportion of their circulating phospholipids in LDL/VLDL (20–30% versus 50% in humans[88]). Moreover, emerging evidence supports the preferential enrichment of DHA in the HDL fraction in human pregnancy, challenging the widely assumed role for VLDL in fetal fatty acid supply[89,90].

Taken together, research from us and others is highlighting how the pregnant state can profoundly contradict many foundational concepts of lipid metabolism. Future work that combines lipidomics with gold-standard isotope-tracing approaches, across multiple

maternal and fetal organs, will likely reveal an extensively changed metabolic system in pregnancy.

## Methods

### Study approval

All animal experiments were approved by the UK Government Home Office under licence P24AB281B awarded to MC.

### Animal model

**Mice.** Unless otherwise stated, all samples were drawn from our previous study, which generated C57BL6/J mice with manipulations to the paternally-inherited imprinted gene, *Dlk1*[4]. In brief, homozygote *Dlk1* knock-outs (Dlk1^tmlSrba; null)[91], heterozygotes lacking the non-functional maternal *Dlk1* allele ("maternal heterozygotes" Dlk1^tmlSrba/+; Mat) and wild-type females (Dlk1^+/+; WT) were generated. Animals were housed in a temperature and humidity-controlled room (21 °C, 55% humidity) with a 12-12 h light-dark cycle. Mice were fed standard chow diet (RM3, Special Diets Service; macronutrient and fatty acid composition of the diet is shown in Supplementary Data Table S13) *ad libitum* and were given access to fresh tap water daily. Following weaning at postnatal 21d, mice were housed in single-sex groups (maximum of five per cage) or occasionally singly housed, except when breeding. In all cohorts mice underwent terminal anesthesia with ~0.8 mg of intra-abdominal pentobarbitol (Dolethal, Vetoquinol) per gram of body weight.

**Experimental cohorts.** Our previous study generated various cohorts of virgin and pregnant mice at 15.5 days post coitum (dpc) with manipulated expression of the DLK1 protein. The 8 cohorts used for the current study were allocated numerical identifiers (groups 1-8) as described in Fig. 1A. WT, Mat and null females were crossed with wild-type males to generate pregnant mice that have normal fetal-derived circulating levels of DLK1 and normal (groups 2 and 8) or null (group 5) local DLK1 expression from maternal tissues. Mat and null females were crossed with null males to generate pregnant mice that lacked fetal-derived circulating DLK1 and had normal (group 7) or null (group 4) local DLK1 expression from maternal tissues. Age-matched WT (group 1), Mat (group 6) and null (group 3) females were used as virgin control groups. Terminal plasma and liver samples were flash frozen in liquid nitrogen and stored at −80 °C.

**LXR cohorts.** LXR-manipulated samples were derived from a cohort of LXR WT and *Lxrab*^−/− (LXR double knockout (DKO)) mice generated by Nikolova and colleagues in 2017[33]. In this study, non-pregnant females and pregnant mice at 7.5, 14.5, and 18.5 dpc were euthanized following a 4-hour fast. Frozen serum, liver and fetal tissue samples were stored at −80 °C and used for the current study.

**Lipidomics.** Lipid extraction, mass spectrometry and data processing techniques were conducted using a recently described high-throughput platform[22] and is summarized here in brief.

**Lipid extraction.** Livers were powdered in liquid nitrogen then homogenized using GCTU solution (6 M guanidinium chloride and 1.5 M thiourea) and a hand-held homogenizer (TissueRuptor II, Qiagen). Liver and plasma aliquots were injected into 96-well plates (2.4 mL/well, glass-coated, Esslab Plate + ™). Internal standards (150 μL, internal standard mixture in methanol, see Supplementary Data Table S14[92]), DMT (500 μL, dichloromethane, methanol (3:1) and triethylammonium chloride (500 mg/L)) and water (500 μL) were added to each of the wells (96-channel pipette). Following agitation (96-channel pipette) and centrifugation (3.2 k × g, 2 min), 20 μL of the organic solution was transferred to a 384-well plate (glass-coated, Esslab Plate + ™) and dried (N$_2$ (g)). The dried films were redissolved (*tert*-butylmethyl ether, 20 μL/well, and then MS-mix, 80 μL/well) and the plate was heat-sealed and run immediately.

**Direct-Infusion mass spectrometry (Untargeted Lipidomics).** All samples were infused into an Exactive Orbitrap (Thermo, Hemel Hampstead, UK), using a TriVersa NanoMate (Advion, Ithaca US) and ionized in the positive mode at 1.2 kV for 72 seconds, followed by the negative ionisation mode at −1.5 kV for 66 s. The analysis was then stopped and the tip was discarded between samples. Samples were run in row order and kept at 15 °C throughout acquisition. The instrument was operated in full-scan mode from *m/z* 150 to 1200 Da.

**Liquid chromatography-mass spectrometry (Targeted Lipidomics).** Targeted lipidomics was performed on liver and plasma samples from 6 mice (from 3 virgin and 3 pregnant groups) and was additionally performed on samples from the separate WT and LXR DKO cohort (liver and serum samples from the groups at 14.5 dpc; liver, serum, placenta and fetal liver samples from the groups at 18.5 dpc).

Chromatographic separation of lipid and triglycerides was achieved using a Waters Acquity UPLC CSH C$_{18}$ (50 mm × 2.1 mm, 1.7 μm) LC-column with a Shimadzu UPLC system (Shimadzu UK Limited, Milton Keynes, UK), at a flow rate of 0.5 mL/min maintained at 55 °C. Mass spectrometry detection was performed on a Thermo Exactive orbitrap mass spectrometer (Thermo Scientific, Hemel Hampstead, UK), which was operated in full scan mode from m/z 100–1800 Da (for singly charged species). To identify lipids, signal peaks were detected for the corresponding accurate mass at the correct retention time and were normalised to the total lipid/glyceride signal for that sample.

Signals data were collected across three runs: the first operating in positive ion and negative ionisation continuous switching mode, the second in a continuous negative ionisation mode switching between CID on and off, and the third was in a continuous positive ionisation mode switching between CID on and off. Runs 2 and 3 were used to determine the fatty acid composition of individual peaks in the chromatogram, thereby identifying the configuration of the lipid isoform(s) present.

**³¹P NMR.** Liver homogenates were combined into virgin (groups 1, 3 and 6) and pregnant (groups 2, 5 and 8) pooled groups to give 5−10 mg of phospholipid per NMR sample. ³¹P NMR was run on the pooled samples using our recently described pipeline[93].

**Data processing.** Raw high-resolution mass-spectrometry data were processed using XCMS (www.bioconductor.org) and Peakpicker v2.0 (an in-house R script[94]). Theoretical lists of known species (by *m/z*) were used for both positive and negative ionisation modes (Supplementary Data Table S15). The correlation of signal intensity to concentration of the variable in liver and plasma QC samples (0.25, 0.50, 1.0) was used to identify which lipid signals were linearly proportional to concentration in the sample type and volume used (threshold for acceptance was a correlation of >0.75). Variables that deviated by more than 9 ppm, had a signal/noise ratio of < 3 and had signals for fewer than 50% of samples were discarded, and zero values were interpreted as not measured. Relative abundance was then calculated by dividing each signal by the sum of signals for that sample and expressed per mille (‰). Final DI-MS datasets included 100 and 212 variables respectively in the positive and negative ionisation modes for plasma, and 141 and 315 variables respectively in the positive and negative ionisation modes for liver. All PC data was validated using signals from both ionisation modes and was presented using the positive ionisation mode data.

**Lipidomics Data Analysis.** Positive and negative DI-MS datasets were analysed independently. Univariate analyses were conducted using

Excel (Office 365) and multivariate analyses (MVA) were conducted using MetaboAnalyst 4.0[95]. Principal component analyses (PCA) were first performed to identify and exclude sample outliers based on 95% confidence intervals. Grouped lipid signals were compared between experimental groups using Two-way ANOVA with Sidak's multiple comparisons. Individual lipid signals were compared between experimental groups using a combination of sparse Partial Least Squares-Discriminant Analyses (sPLS-DA, an unsupervised MVA) and Student's *t*-tests, the p-values adjusted for multiple testing using a Bonferroni correction (0·05/sqrt(n)). Individual variables that passed both statistical tests, in at least two genotype-matched replicate comparisons, were regarded as the most important lipids describing the difference between conditions and were classified as candidate biomarkers (CBMs).

Phospholipid signals of interest were then selected for targeted analysis in the LC-MS/MS dataset to verify their signals and identified their associated fatty acids. $sn-1/sn-2$ fatty acid compositions were annotated using the most abundant isoform identified from the LC-MS/MS dataset. Triglyceride DI-MS signals whose FA residues have 54 or more carbons and 5 or more double bonds were expected to contain a LC-PUFA. Diglyceride DI-MS signals, which represent fragmented triglycerides, were expected to contain a LC-PUFA if they consisted of 38 or more carbons and 5 or more double bonds. For the independent WT and LXR DKO cohort, LC-MS/MS concentrations were compared between experimental groups using Bonferroni-adjusted Student's *t*-tests.

## PUFA metabolites analysis

**Lipid extraction.** Liver samples were crushed with a FastPrep-24 Instrument (MP Biomedical, Fisher scientific SAS, Illkirch, France) in HBSS (Invitrogen, 200 µL) and deuterated internal standard mix (5 µL, 400 ng mL$^{-1}$). After two crush cycles (6.5 ms$^{-1}$, 30 s), an aliquot of the suspension (10 µL) was withdrawn for protein quantification and 0.3 mL of cold methanol added to the remaining material, which was then centrifuged ($1016 \times g$, 15 min, 4 °C). The resulting supernatant was exposed to solid phase extraction using HLB plate (OASIS® HLB mg, 96-well plate, Waters, Saint-Quentin-en-Yvelines, France). Briefly, plates were conditioned with methanol (500 µL) and methanol-water (90:10, *v/v*, 500 µL). Samples were loaded at a flow rate of about 0.5 drop/s and, after complete loading, columns were washed with methanol-water (90:10, *v/v*, 500 µL). The columns were then dried under aspiration and lipids were recovered (methanol, 750 µL). The mixture was dried under a stream of N$_2$ (g) and samples were resuspended in methanol (140 µL) and transferred into a running vial (Macherey-Nagel, Hoerdt, France). The mixture was then dried and resuspended in methanol (10 µL) before injection onto the LC column.

**Liquid Chromatography/Tandem mass spectrometry data collection.** 6-keto-prostaglandin F1 alpha (6kPGF$_{1\alpha}$), thromboxane B2 (TxB$_2$), Prostaglandin E2 (PGE$_2$), 8-iso Prostaglandin A2 (8-isoPGA$_2$), Prostaglandin E3 (PGE$_3$), 15-Deoxy-Δ12,14-prostaglandin J2 (15d-PGJ$_2$), Prostaglandin D2 (PGD$_2$), Lipoxin A4 (LxA4), Lipoxin B4 (LxB4), Resolvin D1 (RvD1), Resolvin D2 (RvD2), Resolvin D5 (RvD5), 7-Maresin 1 (7-Mar1), Leukotriene B4 (LtB$_4$), Leukotriene B5 (LtB$_5$), Protectin Dx (PDx), 18-hydroxyeicosapentaenoic (18-HEPE), 5,6-dihydroxyeicosatetraenoic acid (5,6-DiHETE), 9-hydroxyoctadecadienoic acid (9-HODE), 13-hydroxyoctadecadienoic acid (13-HODE), 15-hydroxyeicosatetraenoic acid (15-HETE), 12-hydroxyeicosatetraenoic acid (12-HETE), 8-hydroxyeicosatetraenoic acid (8-HETE), 5-hydroxyeicosatetraenoic acid (5-HETE), 17-hydroxydocosahexaenoic acid (17-HDoHE), 14-hydroxydocosahexaenoic acid (14-HDoHE), 14,15-epoxyeicosatrienoic acid (14,15-EET), 11,12-epoxyeicosatrienoic acid (11,12-EET), 8,9-epoxyeicosatrienoic acid (8,9-EET), 5,6-epoxyeicosatrienoic acid (5,6-EET), 5-oxoeicosatetraenoic acid (5-oxoETE),

Prostaglandin F2α, (PGF$_{2\alpha}$), 13-Hydroxyoctadecadienoic acid (13oxoODE), 9-hydroxyoctadecadienoic acid (9oxoODE), 9,10-dihydroxy-12-octadecenoic acid (9,10-DiHOME), 12,13-dihydroxy-12-octadecenoic acid (12,13-DiHOME), 9-hydroxy-10,12,15-octadecatrienoic acid (9-HOTrE), 13-hydroxy-9,11,15-octadecatrienoic acid (13-HOTrE), 9,10,13-trihydroxy-11-octadecenoic acid (10-TriHOME) and 9,12,13-trihydroxy-11E-octadecenoic acid (12-TriHOME), 3-hydroxydecanoic acid (C10-3OH), 3-hydroxydodecanoic acid (C12-3OH), 3-hydroxytetradecanoic acid (C$_{14}$−3OH), 3-hydroxyhexadecanoate (C$_{16}$−3OH) and 3-hydroxyoctadecanoic acid (C$_{18}$−3OH) were quantified. All the standards were purchased from Avanti® Polar Lipids (Millipore-SIGMA, St. Quentin Fallavier, France) and from Cayman Chemicals (Bertin Bioreagent, Montigny-le-Bretonneux, France). To simultaneously separate 45 lipids of interest and three deuterated internal standards (5-HETEd8, LxA4d4 and LtB4d4), LC-MS/MS analysis was performed on an ultrahigh-performance liquid chromatography system (UHPLC; Agilent LC1290 Infinity) coupled to an Agilent 6460 triple quadrupole MS (Agilent Technologies) equipped with electrospray ionization operating in negative mode. Reverse-phase UHPLC was performed using a Zorbax SB-C$_{18}$ column (Agilent Technologies) with a gradient elution. The mobile phases consisted of water, acetonitrile (ACN), and formic acid (FA) [75:25:0.1 (*v/v/v*)] (solution A) and ACN and FA [100:0.1 (*v/v*)] (solution B). The linear gradient was as follows: 0% solution B at 0 min, 85% solution B at 8.5 min, 100% solution B at 9.5 min, 100% solution B at 10.5 min, and 0% solution B at 12 min. The flow rate was 0.4 mL/min. The autosampler was set at 5 °C, and the injection volume was 5 µL. Data were acquired in multiple reaction monitoring (MRM) mode with optimized conditions[96].

**Data processing.** Peak detection, integration, and quantitative analysis were performed with MassHunter Quantitative analysis software (Agilent Technologies). For each standard, calibration curves were built using 10 solutions at concentrations ranging from 0.95 to 500 ng/mL. A linear regression with a weight factor of 1/X was applied for each compound. The limit of detection (LOD) and the limit of quantification (LOQ) were determined for the 45 compounds using signal-to-noise ratios. The LOD corresponded to the lowest concentration leading to an S/N value > 3, and LOQ corresponded to the lowest concentration leading to an S/N value > 10. All values less than the LOQ were not considered. The concentration of LxB4, RvD1, RvD2, LTB$_5$, 5,6-DiHETE, 14,15-EET and 11,12-EET did not reach the LOD in our samples. Blank samples were evaluated, and their injection showed no interference (no peak detected), during the analysis.

**PUFA metabolites data analysis.** PUFA metabolites that best distinguish experimental groups were identified following the same CBM Discovery statistical workflow as described above.

## Transcriptional analysis

**Microarray dataset analyses.** We re-analysed a previously published Affymetrix microarray of virgin and pregnant (14.5 dpc) mouse livers ($n = 4$ per group) generated by Quinn and colleagues in 2019[24]. CEL files were downloaded from GEO using accession number GSE121202 and processed for QC and differential expression analysis using the limma/Bioconductor package[97] via the Transcriptome Analysis Console (TAC) software (version 4.0.2; ThermoFisher). Genes were considered significantly differentially expressed if they exhibited at least a 1.25-fold difference in expression with a false discovery rate (FDR, based on the Benjamini–Hochberg procedure) of less than 0.05. Heatmaps were generated with R software packages (version 4.1.1) and volcano plots were generated using the EnhancedVolcano/Bioconductor package (version 3.15). Overlap analysis between upregulated differentially expressed genes and transcription factor targets was performed using the ChEA database in EnrichR[32].

**Liver mRNA expression.** mRNA expression of genes of interest derived from the microarray analysis was measured in liver samples from selected experimental groups (virgin vs pregnancy analysis: groups 1, 2, 3 and 5; DLK1+ vs DLK1- analysis: groups 8, 7, 5 and 4) and in liver samples from the separate WT and LXR DKO cohort. For each analysis, all samples were processed together to avoid batch effects. Frozen liver samples were homogenized in TRIzol™ (Thermo Fisher: Waltham, MA, USA) and kept at −80 °C until RNA extraction following the manufacturer's protocol. RNA samples were treated with DNase I (M0303, New England Biolabs (NEB)) and sodium acetate precipitated. RNA integrity was assessed by the identification of intact 18 S and 28 S rRNA bands after agarose gel electrophoresis and purity was confirmed by 260/280 nm absorbance ratios calculated by a NanoDrop spectrophotometer. Complementary DNA (cDNA) was obtained by reverse transcription (RT) using 1 µg of RNA and Moloney murine leukaemia virus (M-MuLV) reverse transcriptase (M0253, NEB), using the first strand cDNA synthesis standard protocol with random primers (S1330, NEB), 2.5 mM dNTPs (N0446, NEB) and RNase inhibitors (M0314, NEB).

Quantitative real-time PCR (RT-qPCR) assays were performed using SYBR Green qRT-PCR master mix (QuantiNova SYBR Green PCR Kit, Qiagen) with primers designed using the Primer3 software (primer3.ut.ee) or using previously reported sequences[4,98–101] (see Supplementary Data Table S16 for complete list) and ordered from Millipore-SIGMA. cDNA samples were diluted 20x and quantification was performed using the relative standard curve method. For the virgin vs pregnancy analysis, target gene expression was normalized to the average expression of hypoxanthine guanine phosphoribosyltransferase 1 (*Hprt*), α-tubulin and TATA-box binding protein (*Tbp*) and compared between group-pairs by Mann–Whitney U tests. Multiple correlation coefficients with Bonferroni corrected *p*-values were calculated for virgin groups (groups 1 and 3) and pregnant groups (groups 2 and 5) to correlate the expression pattern of multiple genes within each condition. For the DLK1+ vs DLK1- analysis, genes were normalised to the average expression of *Hprt* and α-tubulin. Normalised gene expression was compared between all four groups in the DLK1 analysis by one-way ANOVA with Tukey's multiple comparison post hoc tests. For the WT vs LXR DKO analysis, gene expression was normalized to the average expression of *Hprt*, α-tubulin and *Tbp* and compared by two-way ANOVA with Šídák's multiple comparisons.

### Immunohistochemistry

Fresh livers cut into a ~ 5 mm wide sagittal slice at the level of the gall bladder were fixed with Neutral Buffered Formalin (Millipore-SIGMA, HT501128) overnight at 4 °C and dehydrated through an increasing ethanol series the following day. Samples dehydrated to 100% ethanol were then incubated with Histoclear II (National Diagnostics, HS202) (2 x 2 hr incubations), followed by 2 × 2-hour incubations at 65 °C with Histosec® (1.15161.2504, VWR). 5µm histological sections were cut using a Thermo HM325 microtome, mounted on Menzel-Gläser Superfrost®Plus slides (Thermo Scientific, J1810AMNZ) and used for immunofluorescence (IF).

For double IF, histological sections were rehydrated then antigen unmasking was achieved by incubation in boiling Tris-EDTA buffer pH 9 [10 mM Tris Base, 1 mM EDTA] for 20 mins. Histological sections were incubated overnight at 4 °C with the primary antibodies (rabbit anti-human COX1, ab133319, abcam, 1:200; rat anti-mouse Endomucin, sc-65495 Santa Cruz, 1:200), then incubated with Texas Red-conjugated goat anti-Rat (TI-9400 Vector Labs, 1:300) and goat α-rabbit DyLight® 488 (DI1488, Vector Laboratories, 1:300) for 1 hr at RT, and mounted using VECTASHIELD® Antifade Mounting Medium with DAPI (H-1200-10, Vector Laboratories). Isotype controls for each antibody showed no staining under identical conditions.

### Western blot

Frozen liver samples were each lysed in 200 µl Pierce™ RIPA lysis buffer (Thermo Fisher #89900) with 1% Halt™ protease and phosphatase inhibitor (Thermo Scientific #78442). Protein concentrations were determined using the Pierce™ BCA protein assay kit (Thermo Scientific #23227) and all lysates were diluted to 4 µg/µl in Laemmli buffer (National Diagnostics EC-886-10). Proteins were denatured at 95 °C for 5 min and run on a Bolt™ 4–12% Bis-Tris polyacrylamide gel (Thermo Scientific #NW04125BOX). Proteins were transferred to a nitrocellulose membrane (Invitrogen #PB3310) using the Power Blotter–Semi-dry Transfer System (Invitrogen #PB0013). Membranes were blocked with 5% dairy milk in TBST (100 mM Tris (pH7.5), 400 mM NaCl, 0.05% Tween-20) and incubated in anti-PEMT rabbit polyclonal antibody (1:1000, Bio-techne #NBP1-59580) or anti-α-TUBULIN mouse monoclonal antibody (1:10,000, Millipore-SIGMA #T5168) in 5% dairy milk, overnight at room temperature. Membranes were then washed in TBST and incubated with the secondary HRP-conjugated antibodies: polyclonal goat anti-rabbit (1:1000, P0448, DAKO) or polyclonal goat anti-mouse (1:2000, P0447, DAKO) in 5% dairy milk for 1 h at room temperature. Following washing in TBST, membranes were treated with Pierce™ ECL substrate (Thermo Scientific #32132) and chemiluminescence was detected using the iBright FL1000 Imaging System (Invitrogen).

### Statistical analyses and visualisation

For all comparisons between experimental cohorts, significance was only considered if identified in at least two genotype-matched replicate comparisons. Unless otherwise stated, all statistical tests were performed using the GraphPad Prism Software, version 8.4.3 for Windows (www.graphpad.com). In general, two-tailed Student's t-tests were used to perform pairwise comparisons, whereas One-Way ANOVA was used to compare multiple groups, as detailed in respective Figure legends. All graphs were visualised using R software packages (version 4.1.1).

### Reporting summary

Further information on research design is available in the Nature Portfolio Reporting Summary linked to this article.

## Data availability

All lipidomics datasets generated in this study have been deposited in the EMBL-EBI MetaboLights database under accession code MTBLS9901, https://www.ebi.ac.uk/metabolights/MTBLS9901. The NMR data used in this study are available in the EMBL-EBI BioStudies database under accession code S-BSST651. The processed PUFA metabolite data generated in this study is provided as a source data file. The microarray dataset re-analysed in this study is available on the Gene Expression Omnibus under accession code GSE121202. Source data are provided with this paper.

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

## Acknowledgements
Funding was provided by the Medical Research Council (MRC) Grants MR/L002345/1 (MC), MR/R022836/1 (MC) and MR/J001597/1 (AFS). AK and SF were funded by the BBSRC grant BB/M027252/1. CW is a NIHR Senior Investigator (NIHR200254) and was funded by the Wellcome Trust (#092993). RA was supported by a studentship from the NIHR Biomedical Research Centre at Guy's and St Thomas' NHS Foundation Trust and King's College London. We gratefully acknowledge the Meta-Toul (Toulouse metabolomics & fluxomics facilities, www.metatoul.fr) which is part of the French National Infrastructure for Metabolomics and Fluxomics MetaboHUB. NC received funding from the ANR (agence nationale de la recherche): ANR-18-CE14-0039 and ANR-20-CE14-0011.

## Author contributions
Conceptualization, M.C., A.K., N.C; Methodology, R.A., S. F., M.C., A.K., N.C; Investigation, R.A., S.F., M.A. M.C., S.M., A.L.M.; Writing – Original Draft, R.A., S.F., M.C.; Writing – Review & Editing, R.A., S.F., A.L. M., A.C. F-S., N.C., C.W., A.K., M.C.; Funding Acquisition, M.C., A.K., A.C. F-S., N.C., C.W.; Resources, A.K., C.W., A.C. F-S.; Supervision, M.C., N.C., A.K.

## Competing interests
The authors declare no competing interests.
