## [Peer Review File · Nature Communications]

A co-ordinated transcriptional programme in the maternal liver supplies long-chain polyunsaturated fatty acids to the conceptus using phospholipidsREVIEWER COMMENTS

Reviewer #1 (Remarks to the Author):

This study deals with an important research topic; the supply of maternally derived LC-PUFAs to the fetus. To tackle this aim on a mechanistically level, untargeted lipidomics with transcriptional profiling of healthy healthy and genetically-manipulated DLK1 murine models were introduced. In a previous study authors demonstrated that maternal plasma DLK1 levels are elevated in the catabolic phase of pregnancy, and that the major source of this protein was the conceptus. Of note, genetic modification of this pathway in the mother reduced LC-PUFA accumulation in the fetus.

The methodological set up of this study is appropriate and linked to the research question. However, the conclusions drawn from the results need to be better discussed in the context of the importance of the placenta as another compartment with its own lipoprotein metabolism.

Abstract, line 42: To general and non-specific conclusion, authors should address what they mean with „...new molecular strategies...“ How would this work? Rather to conclude here based on their findings.

Discussion, line 427ff: The authors suggest that maternally derived LC-PUFAs in plasma are made available by phospholipids for the developing fetus in a LXR dependent mechanism. In the LXR DKO groups at late gestational there are FA-differences in the fetal liver (Figure 7L), but no changes of placental EPA and AA levels (Figure 7K). What is the role of the placenta in this concept?

- How do these phospholipids interact with the placenta?
- Uptake and transfer of those PUFAs to the fetus, which signals might be involved to end up in the fetal liver, but does have any effect in the placenta
- The proposed hepatic LC-PUFA-PL transcriptional changes as alternative to LXR-induced mechanisms nicely correlates with LC-PUFAs in the fetal liver, observed in this genetic context. Here again, the role of the placenta is not touched.

It has been shown that placentas from SGA (small gestational age) infants contained lower levels of omega-3 (ALA, EPA, DPA, and DHA) and high omega-6/omega-3 ratios (AA/DHA and LA/ALA), as well as low elongase (Elovl5) and high desaturase (D9Dn7 and D5Dn6) activity as compared to normal infants. How would such changes of PUFAs in the placenta, between different metabolic phenotypes in the mother, fit into your model?

Please add and discuss the role the placenta in context of your findings. Although briefly touched in line 550ff differences between the lipoprotein metabolism between rodent and human, the authors should provide a in depth discussion paragraph on TGs in VLDLs between species and again to what extent is it feasible to translate these findings to human lipoprotein metabolism in pregnancy.

Line487 ff: Role of the placental lipases and lysoPC symporter MFSD2A are discussed. How these well-known mechanisms for FA-uptake into the placenta are transferable to VLDL-mediated availability of triglycerides and LC-PUFA-PCs remains speculative. VLDL is hydrolysed by the hepatic lipase und its activity is regulated by PPARs. Moreover, studies indicate that HF/HCD-induced dysregulation of placental lipid hydrolysis contributes to fetal hepatic lipid accumulation and possibly to fetal overgrowth, at least in mice.

DLK1 (expression and loss of function) and localization in the human versus murine placenta should be addressed and discussed additionally.

Reviewer #2 (Remarks to the Author):

In this study, in order to decipher the maternal adaptations required to provide optimal long chain polyunsaturated fatty acids (LC-PUFAs) to the developing fetus, and, in turn, prevent specific adverse neuronal infant outcomes, Risha Amarsi, Samuel Furse and coworkers conducted a very comprehensive study combining blood and liver lipidomics with liver transcriptional profiling on several independent genetically-murine models. The authors focus on the role of maternal hepatic lipids trafficking pathways in the lipid channel driven by the maternal-placental-fetal unit. This study is highly significant for better understanding of interactions between maternal lipid metabolism during gestation and adequate LC-PUFAs supplies to the conceptus.

The authors used virgin and pregnant mice models previously generated in their lab (8 models described in Figure 1A of genetically-mice with manipulated expression of DLK1) or generated by Nikolova and colleagues (LXR double knockout, DKO mice). Subsequent experiments were focused on dams' plasma and liver analyses of lipid classes (using untargeted DI-MS) and fatty acid composition of phospholipids (using targeted LC-MS/MS). Phospholipids were identified as candidate biomarkers using a stringent statistical pipeline combining multivariate (sparse PLS-DA) and univariate (FDR-corrected Student's t-test) workflow. Maternal hepatic biosynthesis and export of LC-PUFA-phospholipids were investigated by re-analyzing a microarray dataset combined with a targeted ChIP enrichment analysis (ChEA) and a validation of genes of interests using RT-qPCR. The genes of interests were divided into genes involved in limiting steps (PUFA desaturation, PC biosynthesis, lipoprotein assembly), in Kennedy pathway, reverse cholesterol transport and de novo lipogenesis.

Overall, the manuscript represents a set of well conducted experiments that provide novel insights into the complex interplay between maternal hepatic lipid metabolism and the biomagnification of the essential PUFA towards the placental-fetal unit. The key results of this study relate to the activation, in late pregnancy, of LC-PUFAs synthesis, from maternal diet precursors, and their incorporation into phospholipids through a specific transcriptional program regulated by their liver X receptor (LXR) and modulated by maternal production of the Delta-like homologue 1 (DLK1), a major protein of the conceptus, previously identified by the authors, as a putative key regulator of maternal fatty acid metabolism during gestation.

However, I have some minor issues with some of the data and discussion of the literature that should be addressed before publication:

Major comment:

- 1- In the introduction section, the paragraph (lines 64-72) describing PUFA can be considerably shortened however, it would have been more interesting to specify why LC-PUFAs are crucial for central nervous system development across the last trimester of pregnancy and early postnatal months as LCPUFAs are considered essential in fetal life due to low or absent delta 5 and 6 desaturase activity in the placenta and fetus. Moreover, there is no elements concerning the sexual dimorphism in specific adverse neuronal infant outcomes due to low n-3 PUFA supplies, such as males who were reported to be more likely to develop attention deficit/hyperactivity in pregnancy with obesity context.
- 2- Why the sex factor was not considered in the present study due to the putative flexibility in conceptus-maternal interactions that not only could impact secretory factors derived from either the conceptus or endometrium and also could be altered by conceptus sex? The authors could also develop this point in the discussion section.
- 3- In the results section, lines 140-143, have the authors any other information about the mitochondrial function such as beta oxidation which can be accessed through the acylcarnitine profiles of dam's plasma?
- 4- In the results section, lines 320-327, the authors can also investigate directly the phospholipase A2 activity through the ratio Lyso-PC/PC.
- 6- In the discussion section, a putative cross-talk between maternal liver and the microvillous membrane of the syncytiotrophoblast of human placental and its putative involvement in the gradient of LC-PUFA supply to the placenta-fetal unit could be raised, due to the recent literature about fetal sex-dependent placental adaptations in pregnancy complicated by obesity, or diabetes. Indeed fatty acid transfer proteins or enzymes involved in LCPUFA synthesis can be differentially

altered regarding hepatic or placental compartment.

5-Lines 461-462, the authors could also consider the sphingomyelin data in their untargeted lipidomics analysis, as there are also choline donors such as PC- Their findings are also in adequation with the higher efficiency of DHA, provided as a phospholipid, than DHA supplied as part of a triglyceride, for brain DHA accretion and infant growth, as reported in literature.

6-Lines 504-512, the high levels in hepatic ARA-enriched phospholipids raises the role of oxylipins and COX/LOX pathways in pregnancy, and notably pregnancy complicated by obesity, preeclampsia...Could the authors discuss this point and have they arguments for specific inflammatory environment associated to their genetically mice models?

The figures and Tables are very clear and the legends provide full of details. We appreciate the study design and schematic depicting lipid biosynthesis pathways and/or the transcriptomic analysis

1- In the title, please develop LC-PUFA

2- P2-. In the abstract section, replace "untargeted lipidomics" by "a comprehensive approach combining untargeted and targeted lipidomics with" and specify which "genetically-manipulated murine models" were used.

3- In the Methods section, lines 770-777, the authors reported animal study in accordance with the ARRIVE (Animal Research: Reporting of In Vivo Experiments) guidelines. Please add the details of energy supply for each macro-nutrients in the standard chow diet (Supplementary Table S12).

4- In Supplementary Table S6, undetermined means under LOD ??

5- Concerning the sentence lines 788_790, "Age-matched WT (group 1), Mat (group 6) and null (group 3) females were used as virgin control groups", it would be interesting to know if pregnant females presented similar pre-gravid body weight as those of virgin females?

6- Lines 982, please develop the abbreviations Hprt and Tbp

7- In the statistical section, lines 1028-1031, it would be appreciable to detail the different tests used to compare the groups as reported in figures' legends.

8- Line 1032 Please add the agreement of the animal study in the study approval section

Reviewer #3 (Remarks to the Author):

This study extends the authors' previous study (2016 Cleaton MAM et al.), and the authors conducted lipidomics, PUFA metabolite analysis, and Transcriptional analysis using stored samples derived from various cohorts of mouse models with different genotypes. The authors demonstrated a late pregnancy-specific, selective activation of the Liver X Receptor signaling pathway which dramatically increases maternal supply of LC-PUFAs within circulating phospholipids. Although this study applied new analytical methods and showed interesting findings, the reviewers have several comments and some concerns.

Major comment 1

The authors previously stated that Fetus-derived DLK1 is required for maternal metabolic adaptations to pregnancy (2006 Cleaton MM et al.). However, the present study showed that the hepatic transcriptional program is co-ordinately regulated by the liver X receptor (LXR) and modulated by maternal production of DLK1. The reviewer could not understand how to interpret the conclusions from the previous study and the findings in the present study. Which is critical DLK1 from the mother or fetus? The reviewer considers that a more careful explanation is needed.

Major comment 2

The authors described as follows; We observed that loss of DLK1 in the fetus caused impairments in maternal fasting metabolism and lipoprotein production, whereas loss of DLK1 in the dam prevented the normal acquisition and release of her adipose tissue stores (lines 59-62). The authors need to describe what is known and what is unknown about how DLK1 affects fatty acid metabolism in pregnant mothers. Furthermore, the authors need to show the known information in the introduction section on what kind of fatty acids are affected by DLK1 in pregnant females.

Major comment 3

The mother may not produce DLK1 but the fetus may have DLK1 production (G5), or the mother may produce DLK1 but the fetus may produce DLK1 (G7). Reviewers would like to know whether the total amount of DLK1 in late pregnancy is important or whether DLK1 production differs depending on whether it originates from the mother or fetus.

Major comment 4

The authors compared various lipids fatty acids levels and transcriptional levels of targeted genes in G3 and G5 as well as in G1 and G2 and in G6 and G8. G3 and G5 have no DLK1 from maternal own origin, while G5 has circulating DLK1 from fetal origin. The reviewer would like to know whether the circulating DLK1 levels in G3 and G5 are the same or different from those in G1, G2, G6, and G8.

Major comment 5

The authors compare maternal circulating and hepatic lipid profiles among the mice models with different maternal genotypes WT, Mat (DLK1+), and Null (DLK1-) between non-pregnancy and pregnancy, respectively (Figures 2A-H). Similar changes are observed in each genotype between non-pregnancy and pregnancy, but TG (Liver) is not different between non-pregnancy and pregnancy in the null (DLK1-) mice model (Figure 2B). Does the absence of DLK1 derived from the mother have anything to do with this?

Major comment 6

The authors described as following: Despite the influence of DLK1 on hepatic PE(18:0/20:4), no effect was observed on PUFA metabolite concentration (lines 264-265). Could you show the related data?

Minor comment 1

In line 27, the authors cited Figure 3B. Is it a mistake of Figure 5B?

REPLY TO REVIEWERS' COMMENTS

We thank the reviewers for their overall detailed and positive response to our manuscript. By addressing their critique we have significantly improved the clarity of the manuscript, and included important discussion points which were not addressed previously. A detailed response to each comment is included below. Line numbers refer to the ordering in the manuscript version with tracked changes.

Reviewer #1 (Remarks to the Author):

This study deals with an important research topic; the supply of maternally derived LC-PUFAs to the fetus. To tackle this aim on a mechanistically level, untargeted lipidomics with transcriptional profiling of healthy healthy and genetically-manipulated DLK1 murine models were introduced.

In a previous study authors demonstrated that maternal plasma DLK1 levels are elevated in the catabolic phase of pregnancy, and that the major source of this protein was the conceptus. Of note, genetic modification of this pathway in the mother reduced LC-PUFA accumulation in the fetus.

The methodological set up of this study is appropriate and linked to the research question. However, the conclusions drawn from the results need to be better discussed in the context of the importance of the placenta as another compartment with its own lipoprotein metabolism.

We thank the reviewer for their positive comments.

Abstract, line 42: To general and non-specific conclusion, authors should address what they mean with „...new molecular strategies...“ How would this work? Rather to conclude here based on their findings.

We have edited the abstract to include these suggestions (L31-48).

Discussion, line 427ff: The authors suggest that maternally derived LC-PUFAs in plasma are made available by phospholipids for the developing fetus in a LXR dependent mechanism. In the LXR DKO groups at late gestational there are FA-differences in the fetal liver (Figure 7L), but no changes of placental EPA and AA levels (Figure 7K). What is the role of the placenta in this concept?

- How do these phospholipids interact with the placenta?
- Uptake and transfer of those PUFAs to the fetus, which signals might be involved to end up in the fetal liver, but does have any effect in the placenta
- The proposed hepatic LC-PUFA-PL transcriptional changes as alternative to LXR-induced mechanisms nicely correlates with LC-PUFAs in the fetal liver, observed in this genetic context. Here again, the role of the placenta is not touched.

We agree that this is an important point, and we were initially surprised to find that maternal LXR \$\alpha\$ / \$\beta\$ deletion (LXR-DKO) resulted in reduction in LC-PUFA accumulation in the fetal liver at 18.5 dpc without apparent alteration to the levels of these FAs in the placenta.

The transfer of LC-PUFAs from mother to fetal tissue must be considered as a multi-step process. Firstly, the FAs must be modified, packaged and mobilised by maternal tissues for transfer to the maternal plasma. Second, the FAs must be taken up by the trophoblast cells, potentially modified by placental metabolic enzymes, and transported across the placental

barrier. Finally, the FAs must be transported into the fetal circulation, then incorporated into fetal tissues. The steady-state level of placental LC-PUFA levels will reflect several aspects of this process – maternal concentrations of circulating lipids, rate of uptake, accumulation into the placental metabolic pool and placental export.

The experiment using the LXR-DKO described in our manuscript functionally manipulates only the first part of this process – maternal supply of lipids to the fetus. Here we demonstrate clearly that LXR α/β function in the dam is necessary for ~50% of the total accumulation of fetal liver phospholipid (PL) associated DHA and arachidonic acid (ARA).

Without additional tracing experiments we cannot definitively say how the rate of placental transfer is modified in response to low maternal PL-LC-PUFA supply (stated as a limitation in our discussion). However, the lack of concordance between the placental and fetal compartment may be explained in several ways:

Firstly, we see higher overall concentration and a bigger differential between wild-type and LXR-DKO levels of plasma PL-DHA and AA at 14.5 dpc compared to 18.5 dpc (Figure 7J). This is consistent with the greater induction of expression of LXR target genes at the 14.5 dpc time-point (Figure 7B-H). We measured lipids in fetus and placenta at the end of gestation (18.5 dpc, earlier samples were not available). This stage represents the sum total of PL-DHA/AA accumulation in the fetus over the course of pregnancy, but not the steady state level in the placenta. Future work will employ stable isotope labelling methodologies to resolve this issue.

Secondly, recent studies employing mathematical models in combination with stable isotope tracing of FAs in a human placental perfusion system have demonstrated that the placenta accumulates a relatively low level of DHA compared to other FAs (e.g, roughly 10-fold less compared to FA (16:0), PMID: 33637949). This is in agreement with historical findings that DHA is rapidly transported across the placenta with little placental accumulation. Interestingly an opposite effect is observed for n-6 PUFAs, particularly ARA which does accumulate in the placenta, possibly in reserve for prostanoid and endocannabinoid synthesis (PMID: 7277037, PMID: 32347309). In keeping with these findings, in our data PC-associated DHA in the placenta reached only 50% of the concentration in the fetal liver, replicating biomagnification (Figure 7K). Moreover, placental PC-ARA levels appeared to exceed those in the maternal plasma and fetal liver suggesting placental accumulation.

It has been shown that placentas from SGA (small gestational age) infants contained lower levels of omega-3 (ALA, EPA, DPA, and DHA) and high omega-6/omega-3 ratios (AA/DHA and LA/ALA), as well as low elongase (Elovl5) and high desaturase (D9Dn7 and D5Dn6) activity as compared to normal infants. How would such changes of PUFAs in the placenta, between different metabolic phenotypes in the mother, fit into your model?

How specific lipids and their FAs are partitioned to the placental metabolic pool or the fetal circulation is an extremely important topic, since these processes are modified by maternal diabetes and obesity (PMID: 37288271, PMID: 33637949). There is also an increasing appreciation that fetal/placental sex interacts with maternal diet to modulate placental adaptation to compromised maternal metabolic health (PMID: 32717842, 32452400 and see response to reviewer 2 below). Future studies where placental, fetal and maternal lipid metabolism are manipulated in isolation will be required to fully understand these important mechanisms.

Please add and discuss the role the placenta in context of your findings.

We have added the points detailed above to the discussion section (L566-591).

Although briefly touched in line 550ff differences between the lipoprotein metabolism between rodent and human, the authors should provide a in depth discussion paragraph on TGs in VLDLs between species and again to what extent is it feasible to translate these findings to human lipoprotein metabolism in pregnancy.

We agree that caution should be applied when extrapolating results from rodent pregnancy to other species, including humans. Differences between mouse and human lipoprotein metabolism have been underscored by many decades of work by investigators attempting to phenocopy the effects of cholesterol metabolism on cardiovascular disease. In this case it is clear that mice do not spontaneously generate atherosclerotic plaques, due at least in part by their low VLDL/LDL cholesterol levels compared to HDL-C, and lack of the enzyme CETP. In contrast to knowledge about cholesterol metabolism, much less is appreciated about mechanisms and species differences in lipid transport in the lipoprotein compartment. Although they are known to be 'HDL dominant', mice carry a significant proportion of their circulating phospholipids in LDL/VLDL (20-30% versus 50% in humans, PMID: 25894274).

While the widespread use of classical biochemical markers mean that levels of total cholesterol, bulk TG and HDL-C (and inferred measurements of LDL-C) are often reported in both human and rodent pregnancy studies, the reviewers will appreciate that details of the maternal lipidome are much sparser. While changes in the density, cholesterol and TG content have been described (PMID: 11134106), to our knowledge the FA composition in TG in different lipoprotein particle classes has not yet been fully described in pregnancy.

As the reviewer requests we have expanded the limitations section of the discussion to include some of these points, L613-625

Line487 ff: Role of the placental lipases and lysoPC symporter MFSD2A are discussed. How these well-known mechanisms for FA-uptake into the placenta are transferable to VLDL-mediated availability of triglycerides and LC-PUFA-PCs remains speculative. VLDL is hydrolysed by the hepatic lipase und its activity is regulated by PPARs. Moreover, studies indicate that HF/HCD-induced dysregulation of placental lipid hydrolysis contributes to fetal hepatic lipid accumulation and possibly to fetal overgrowth, at least in mice.

In this paper we show a generalised increase in circulating TGs in pregnancy. This rise is known to occur due to increased TG into all lipoprotein types (PMID: 34416270), but especially VLDL, due to reduced maternal tissue expression of lipoprotein lipase (LPL), hepatic lipase (HL) and increased expression of cholesterol ester transport protein (CETP). While rodents do not have CETP, the drop in adipose LPL is well reported (PMID: 5528598), and we saw a significant reduction in hepatic expression of HL (*Lipc*, Figure 5I). We believe that this increase in circulating TG in lipoproteins is the source of fatty acids of all types to the placenta. When incorporated into TG in VLDL, FAs are taken up into the placenta by a variety of mechanisms including VLDL-R and placental lipase-FATP-mediated routes.

Our data suggests that in addition to the elevation of all FAs in the maternal plasma (in TG), there is a specific, LXR-induced enrichment for LC-PUFAs in phospholipids. It is well known that endothelial lipase is the major lipase found on the maternal blood-facing surface of trophoblast cells in the human placenta (PMID: 17356047). EL is primarily a phospholipase A2 with high affinity for HDL, thus predicted to release *sn-1* FAs (PMID: 10192396). In contrast, lipoprotein lipase, the major TG lipase, is expressed at lower levels than EL in villous cytotrophoblasts and villous syncytiotrophoblast cells in early pregnancy and is not present at term (PMID: 17356047). Mice express high levels of both lipases in the labyrinthine zone of the placenta, but to our knowledge the cellular localisation has not been

reported (PMID: 16150822). Interestingly, HDL in pregnancy is enriched for phospholipid LC-PUFAs compared to the other lipoproteins, suggesting a relatively unexplored role for HDL in gestational FA supply (PMID: 30670033). Overall, these data suggest that a major route for LC-PUFA transfer from lipoproteins is by EL-mediated cleavage of *sn-1* FAs from PLs, and potentially transfer of LPC-*sn2* lipids by MFSD2A, as noted previously by others and discussed by us in lines 503-524. We have expanded the discussion to clarify this point.

DLK1 (expression and loss of function) and localization in the human versus murine placenta should be addressed and discussed additionally.

DLK1 is highly expressed in the human and mouse placenta, and the expression pattern is very similar between species. Some ambiguity in the literature exists about this similarity, possibly because DLK1 is a secreted protein that may be detected in locations distal from its cellular source. This interpretation is based on substantial unpublished work from our group, including the use of high quality single-cell/nuclear sequencing datasets that have become available over the past few years.

'Figure and Corresponding Legend Redacted'

In the mouse, *Dlk1* is expressed in the extraembryonic mesoderm-derived (EEM) cells of the placenta, and there is no contribution of mRNA from the trophoblast. We observe expression from midgestation (e9.5) in the allantoic cells invading the chorion, and subsequently in fetal endothelial cells and mesenchyme derived from the EEM (Figure 1a). This expression pattern is confirmed by single-nuclear sequencing of the mouse placenta from e9.5-e14.5 (PMID: 33141023), where *Dlk1* mRNA is confined to the same cell populations and is absent from the trophoblast (Figure 1b). We have previously reported DLK1 immunostaining in the mouse placenta where the *Dlk1* gene had been deleted from the EEM using conditional targeting (PMID: 27776119). In this case we identified a small number of putative trophoblast cells which were positive for DLK1. We suggest that this protein may have been in transit across the trophoblast layer and was not directly synthesised by these cells, since we have never observed *Dlk1* in trophoblast using methods for mRNA detection.

In humans we see similar localisation of DLK1. In single cell data from the second trimester human placenta (GW17-24, PMID: 35796428) we can detect no *DLK1* transcripts produced from the trophoblast, instead there is very high expression in the EEM-derived mesenchyme of the villi, and a small amount in the fetal endothelial cells (Figure 1c). Similar to our work in the mouse, others have detected DLK1 protein in the syncytiotrophoblast layer (PMID: 26242929) – but to our knowledge mRNA expression has not been confirmed in these cells.

We propose that these data are not relevant to this manuscript. They will form part of a substantive paper about the role of DLK1 in placental function that we are currently preparing for publication. However, we have included more detail about the *Dlk1*-deletion model -please see our reply to Reviewer 3.

Reviewer #2 (Remarks to the Author):

In this study, in order to decipher the maternal adaptations required to provide optimal long chain polyunsaturated fatty acids (LC-PUFAs) to the developing fetus, and, in turn, prevent specific adverse neuronal infant outcomes, Risha Amarsi, Samuel Furse and coworkers conducted a very comprehensive study combining blood and liver lipidomics with liver transcriptional profiling on several independent genetically-murine models. The authors focus on the role of maternal hepatic lipids trafficking pathways in the lipid channel driven by the maternal-placental-fetal unit. This study is highly significant for better understanding of interactions between maternal lipid metabolism during gestation and adequate LC-PUFAS

supplies to the conceptus.

The authors used virgin and pregnant mice models previously generated in their lab (8 models described in Figure 1A of genetically-mice with manipulated expression of DLK1) or generated by Nikolova and colleagues (LXR double knockout, DKO mice). Subsequent experiments were focused on dams' plasma and liver analyses of lipid classes (using untargeted DI-MS) and fatty acid composition of phospholipids (using targeted LC-MS/MS). Phospholipids were identified as candidate biomarkers using a stringent statistical pipeline combining multivariate (sparse PLS-DA) and univariate (FDR-corrected Student's t-test) workflow. Maternal hepatic biosynthesis and export of LC-PUFA-phospholipids were investigated by re-analyzing a microarray dataset combined with a targeted ChIP enrichment analysis (ChEA) and a validation of genes of interests using RT-qPCR. The genes of interests were divided into genes involved in limiting steps (PUFA desaturation, PC biosynthesis, lipoprotein assembly), in Kennedy pathway, reverse cholesterol transport and de novo lipogenesis.

Overall, the manuscript represents a set of well conducted experiments that provide novel insights into the complex interplay between maternal hepatic lipid metabolism and the biomagnification of the essential PUFA towards the placental-fetal unit. The key results of this study relate to the activation, in late pregnancy, of LC-PUFAs synthesis, from maternal diet precursors, and their incorporation into phospholipids through a specific transcriptional program regulated by their liver X receptor (LXR) and modulated by maternal production of the Delta-like homologue 1 (DLK1), a major protein of the conceptus, previously identified by the authors, as a putative key regulator of maternal fatty acid metabolism during gestation. However, I have some minor issues with some of the data and discussion of the literature that should be addressed before publication:

Major comment:

1- In the introduction section, the paragraph (lines 64-72) describing PUFA can be considerably shortened however, it would have been more interesting to specify why LC-PUFAs are crucial for central nervous system development across the last trimester of pregnancy and early postnatal months as LCPUFAs are considered essential in fetal life due to low or absent delta 5 and 6 desaturase activity in the placenta and fetus. Moreover, there is no elements concerning the sexual dimorphism in specific adverse neuronal infant outcomes due to low n-3 PUFA supplies, such as males who were reported to be more likely to develop attention deficit/hyperactivity in pregnancy with obesity context.

We have made some edits to this section to prioritise the information (L70-97). We agree that there is strong evidence for low and absent delta 5 and 6 desaturase activity in the fetus (we cite PMID: 7285840 in the manuscript). However, there is some evidence supporting placental production of these enzymes in rodents and humans (for example PMID: 32717842, 32452400). For this reason we have not modified the sentence "Fetal synthesis can only account for a small proportion of this demand, and instead maternal production/mobilisation and placental transfer is required to meet the ARA and DHA requirements for healthy development" (L79-81).

2- Why the sex factor was not considered in the present study due to the putative flexibility in conceptus-maternal interactions that not only could impact secretory factors derived from either the conceptus or endometrium and also could be altered by conceptus sex? The authors could also develop this point in the discussion section.

We are aware of the important fetal sex differences in both placental responses to LC-PUFA uptake and developmental outcome in response to maternal obesity described by Powell et. al. and others (PMID: 33321178, PMID: 37770949). However, our study focuses primarily on the maternal hepatic mechanism of biomagnification rather than the fetal/placental uptake route. Because rodent pregnancies contain multiple pups of mixed sex, it is not possible to dissect the impact of fetal sex on the maternal hepatic response in our study (as it is likely to be an average of both sexes). Our future work can address both placental compensatory mechanisms of reduced maternal PUFA supply (as mentioned in the response to Reviewer 1, above) and should include offspring sex in this analysis. Consequently, this is outside the scope of the current work, which is already a complex story, and so we have not included these points in the Introduction. However, since the issue of fetal sex is an important one, we have included the following sentence in the limitations section of the Discussion; “Recent work has highlighted an important role for fetal sex in placental metabolism and transfer of DHA in the context of maternal obesity. Specifically, male offspring of obese mothers have reduced placental and plasma levels of DHA compared to females, or the offspring of lean mothers (PMID: 33321178, PMID: 37770949). Our study focuses on the maternal hepatic mechanism of biomagnification rather than the fetal/placental uptake route. Because rodent pregnancies contain multiple pups of mixed sex, it is not possible to dissect the impact of fetal sex on the maternal hepatic response. Future work must address both placental compensatory mechanisms of reduced maternal LC-PUFA supply and should include offspring sex in this analysis.” (L604-612).

3- In the results section, lines 140-143, have the authors any other information about the mitochondrial function such as beta oxidation which can be accessed through the acylcarnitine profiles of dam’s plasma?

We did not measure acylcarnitine profiles in these mice, as this class of metabolites is not covered by the lipidomics method. Since the animals were free-fed we did not expect to observe an appreciable level of mitochondrial beta-oxidation. Nevertheless, as part of our bioactive lipids panel we measured hepatic 3-hydroxy-medium and long-chain fatty acids. We observed no differences between groups (Figure 4A).

4- In the results section, lines 320-327, the authors can also investigate directly the phospholipase A2 activity through the ratio Lyso-PC/PC.

We attempted to use the LPC(16:0)/PC(36:4) and LPC(18:0)/PC(38:4) ratios in the liver as a proxy to determine if PLA₂ activity was increased in pregnancy. We saw the opposite relationship. We suggest that this signal was swamped by the large increase in LC-PUFA synthesis at this time, making the PLA₂ effect hard to isolate. A good estimation of PLA₂ activity must assume similar rates of synthesis between conditions, which is not the case in our data.

6- In the discussion section, a putative cross-talk between maternal liver and the microvillous membrane of the syncytiotrophoblast of human placental and its putative involvement in the gradient of LC-PUFA supply to the placenta-fetal unit could be raised, due to the recent literature about fetal sex-dependent placental adaptations in pregnancy complicated by obesity, or diabetes. Indeed fatty acid transfer proteins or enzymes involved in LCPUFA synthesis can be differentially altered regarding hepatic or placental compartment.

We have expanded the Discussion section to include a section about the complexity of the interplay between maternal, fetal and placental compartments, including the role of fetal sex

in placental adaptations to compromised maternal metabolism; obesity and diabetes (L566-593).

5-Lines 461-462, the authors could also consider the sphingomyelin data in their untargeted lipidomics analysis, as there are also choline donors such as PC- Their findings are also in adequation with the higher efficiency of DHA, provided as a phospholipid, than DHA supplied as part of a triglyceride, for brain DHA accretion and infant growth, as reported in literature.

The sphingomyelin (SM) data is shown in Tables S1 and S2. Since SMs were not altered in pregnancy as a class, and only 2 SMs were found to show differential abundance in our candidate biomarker analysis (Table S4), SM(36:1) (up in pregnant liver) and SM(33:1) (up in pregnant plasma), we have chosen not to highlight these results in an already complex paper.

We have not included the second point as it appears to be unresolved in the literature. Indeed, a recent substantive review of fetal DHA accretion according to lipid source stated that TGs and PLs perform equally well with current levels of supplementation, but concluded that more data is required to conclusively determine this point (PMID: 33557158).

6-Lines 504-512, the high levels in hepatic ARA-enriched phospholipids raises the role of oxylipins and COX/LOX pathways in pregnancy, and notably pregnancy complicated by obesity, preeclampsia...Could the authors discuss this point and have they arguments for specific inflammatory environment associated to their genetically mice models?

We agree that the contribution of prostanoid synthesis to pregnancy complications is an exciting topic that is highlighted by our analysis. To our knowledge ours is the first unbiased screen of hepatic bioactive lipids in pregnancy, so much work will need to be done to follow up these initial findings. Our animals are housed in a 'clean' but not sterile facility, thus we do not expect the animals to be exposed to a specific inflammatory environment. Since we feel it is too soon to speculate about the relationship between specific COX-derived lipid mediators and pregnancy complications, we wish to keep the current wording of this section as "Given the reported associations between prostanoids and pre-eclampsia (70–72), and their critical regenerative roles following hepatic injury (73), further research into maternal hepatic bioactive PUFA metabolites may elucidate underlying mechanisms that implicate liver dysfunctions to pregnancy complications. " (L534-538).

The figures and Tables are very clear and the legends provide full of details. We appreciate the study design and schematic depicting lipid biosynthesis pathways and/or the transcriptomic analysis.

We thank the reviewer for their extremely helpful comments throughout.

1- In the title, please develop LC-PUFA

This has been done (L2).

2- P2-. In the abstract section, replace "untargeted lipidomics" by "a comprehensive approach combining untargeted and targeted lipidomics with" and specify which "genetically-manipulated murine models" were used.

We have edited the abstract to include these suggestions (L31-48).

3- In the Methods section, lines 770-777, the authors reported animal study in accordance with the ARRIVE (Animal Research: Reporting of In Vivo Experiments) guidelines. Please add the details of energy supply for each macro-nutrients in the standard chow diet (Supplementary Table S12).

These values have been added to Table S13.

4- In Supplementary Table S6, undetermined means under LOD ??

Yes, the table legend has been amended to make this clear.

5- Concerning the sentence lines 788_790, "Age-matched WT (group 1), Mat (group 6) and null (group 3) females were used as virgin control groups", it would be interesting to know if pregnant females presented similar pre-gravid body weight as those of virgin females?

Body mass did not differ between groups 1, 6 and 3. However, adipose mass was significantly increased in group 3 (abdominal white adipose tissue mass = 0.28g in group 3 compared to 0.18g in group 1 and 0.17g in group 6). This is detailed in our previous paper (PMID: 27776119). We have modified the discussion to more clearly articulate this. (L546-551)

6- Lines 982, please develop the abbreviations Hprt and Tbp

This has been done.

7- In the statistical section, lines 1028-1031, it would be appreciable to detail the different tests used to compare the groups as reported in figures' legends.

We have added: In general, two-tailed Student's t-tests were used to perform pairwise comparisons, whereas One-Way ANOVA was used to compare multiple groups, as detailed in respective Figure legends. (L1110-1112).

8- Line 1032 Please add the agreement of the animal study in the study approval section

This has been added.

Reviewer #3 (Remarks to the Author):

This study extends the authors' previous study (2016 Cleaton MAM et al.), and the authors conducted lipidomics, PUFA metabolite analysis, and Transcriptional analysis using stored samples derived from various cohorts of mouse models with different genotypes. The authors demonstrated a late pregnancy-specific, selective activation of the Liver X Receptor signaling pathway which dramatically increases maternal supply of LC-PUFAs within circulating phospholipids. Although this study applied new analytical methods and showed interesting findings, the reviewers have several comments and some concerns.

Major comment 1

The authors previously stated that Fetus-derived DLK1 is required for maternal metabolic adaptations to pregnancy (2006 Cleaton MM et al.). However, the present study showed that

the hepatic transcriptional program is co-ordinately regulated by the liver X receptor (LXR) and modulated by maternal production of DLK1. The reviewer could not understand how to interpret the conclusions from the previous study and the findings in the present study. Which is critical DLK1 from the mother or fetus? The reviewer considers that a more careful explanation is needed.

We hope that the concerns expressed in this comment are dealt with by the changes described below, and in modifications suggested by the other reviewers.

Major comment 2

The authors described as follows; We observed that loss of DLK1 in the fetus caused impairments in maternal fasting metabolism and lipoprotein production, whereas loss of DLK1 in the dam prevented the normal acquisition and release of her adipose tissue stores (lines 59-62). The authors need to describe what is known and what is unknown about how DLK1 affects fatty acid metabolism in pregnant mothers. Furthermore, the authors need to show the known information in the introduction section on what kind of fatty acids are affected by DLK1 in pregnant females.

Prior to this work the unbiased lipid profile of the pregnant mouse in late gestation had not (to our knowledge) been reported. In addition, we had no information about changes to the lipid species associated with DLK1 modification in pregnancy. In the Cleaton 2016 (PMID: 27776119) paper we had described the lipid pathways in only broad brushstrokes – measuring serum cholesterol (including HDL-cholesterol), TAG, unesterified fatty acids and a major ketone species (3-hydroxy-butyrate). In addition we had measured maternal weight gain and body composition.

To make it clear to the reader that this was our starting point, we have modified the section highlighted by the reviewer: “This work led us to hypothesise that DLK1 is a key modulator of maternal fatty acid metabolism in pregnancy, yet a detailed examination of pregnancy-associated lipid species has not been performed.” (L67-69).

Major comment 3

The mother may not produce DLK1 but the fetus may have DLK1 production (G5), or the mother may produce DLK1 but the fetus may produce DLK1 (G7). Reviewers would like to know whether the total amount of DLK1 in late pregnancy is important or whether DLK1 production differs depending on whether it originates from the mother or fetus.

In adult female mice plasma DLK1 levels are low (<60ng/mL). However, in late gestation these levels rise 5x to 250-300ng/mL in the maternal plasma. Fetal DLK1 is the source of this DLK1 in the maternal circulation (PMID: 27776119). In terms of the sum maternal plasma DLK1 in pregnancy, the major contribution is thus of fetal, rather than maternal tissue origin. To clarify this point we have added the previously measured mean plasma DLK1 levels by experimental group to the schematic in Figure 1A.

However, in this study we found that it is the maternal genotype that contributes most to the maternal mobilisation of PUFAs. To clarify further we added the following sentence to the Discussion: “In our previous study we found that fetally-derived DLK1 in maternal circulation had a minimal impact on fed metabolism (4). Consistently, here fetal DLK1 led to a small shift in total PE:PC ratios (Figure 2I, J). However, the ablation of *Dlk1* in maternal tissues modified the dynamics of lipid mobilisation, potentially via indirect effects on whole body adipose storage.”

Major comment 4

The authors compared various lipids fatty acids levels and transcriptional levels of targeted genes in G3 and G5 as well as in G1 and G2 and in G6 and G8. G3 and G5 have no DLK1 from maternal own origin, while G5 has circulating DLK1 from fetal origin. The reviewer would like to know whether the circulating DLK1 levels in G3 and G5 are the same or different from those in G1, G2, G6, and G8.

Please see the comment above.

Major comment 5

The authors compare maternal circulating and hepatic lipid profiles among the mice models with different maternal genotypes WT, Mat (DLK1+), and Null (DLK1-) between non-pregnancy and pregnancy, respectively (Figures 2A-H). Similar changes are observed in each genotype between non-pregnancy and pregnancy, but TG (Liver) is not different between non-pregnancy and pregnancy in the null (DLK1-) mice model (Figure 2B). Does the absence of DLK1 derived from the mother have anything to do with this?

We agree, and have modified part of the discussion “We found that dams lacking DLK1 derived from maternal tissues had modestly increased transcription of rate-limiting steps in the hepatic synthesis and export of LC-PUFA-phospholipids. Notably, in our previous study, these mice commenced pregnancy with a higher adipose mass, but exhibited a reduced ability to gain adipose tissue during pregnancy (PMID: 27776119), and appear to accumulate less hepatic TG (Figure 2B).” L546-551.

Major comment 6

The authors described as following: Despite the influence of DLK1 on hepatic PE(18:0/20:4), no effect was observed on PUFA metabolite concentration (lines 264-265). Could you show the related data?

All data from the PUFA metabolite screen is now shown in Supplementary Table S8.

Minor comment 1

In line 27, the authors cited Figure 3B. Is it a mistake of Figure 5B ?

Changed, thanks.

REVIEWERS' COMMENTS

Reviewer #1 (Remarks to the Author):

The authors responded to all my queries comprehensively and in details. Strengths and limitations are now clearly explained and understandable. Thank you

Reviewer #2 (Remarks to the Author):

The authors have provided detailed, well-argued responses to previous comments. The soundness of the present research are of high quality, given the complexity of the field of research on the interaction between maternal, fetal and placental compartments.

Reviewer #3 (Remarks to the Author):

The authors have responded appropriately to the reviewers' comments. However, regarding the response to the major comment 3, the reviewer was not able to see the description of plasma DLK1 concentration in Figure 1A in the revised manuscript. Please add the DLK1 concentrations to the Figure 1A.

REPLY TO REVIEWERS' COMMENTS

Revised version comments

Reviewer #1 (Remarks to the Author):

The authors responded to all my queries comprehensively and in details. Strengths and limitations are now clearly explained and understandable. Thank you

Reviewer #2 (Remarks to the Author):

The authors have provided detailed, well-argued responses to previous comments. The soundness of the present research are of high quality, given the complexity of the field of research on the interaction between maternal, fetal and placental compartments.

Reviewer #3 (Remarks to the Author):

The authors have responded appropriately to the reviewers' comments. However, regarding the response to the major comment 3, the reviewer was not able to see the description of plasma DLK1 concentration in Figure 1A in the revised manuscript. Please add the DLK1 concentrations to the Figure 1A.

Response: Apologies for not including the plasma DLK1 concentrations in Figure 1A in the revised manuscript. The concentrations have now been added to the figure.

Version 1 comments

We thank the reviewers for their overall detailed and positive response to our manuscript. By addressing their critique we have significantly improved the clarity of the manuscript, and included important discussion points which were not addressed previously. A detailed response to each comment is included below. Line numbers refer to the ordering in the manuscript version with tracked changes.

Reviewer #1 (Remarks to the Author):

This study deals with an important research topic; the supply of maternally derived LC-PUFAs to the fetus. To tackle this aim on a mechanistically level, untargeted lipidomics with transcriptional profiling of healthy healthy and genetically-manipulated DLK1 murine models were introduced.

In a previous study authors demonstrated that maternal plasma DLK1 levels are elevated in the catabolic phase of pregnancy, and that the major source of this protein was the conceptus. Of note, genetic modification of this pathway in the mother reduced LC-PUFA accumulation in the fetus.

The methodological set up of this study is appropriate and linked to the research question. However, the conclusions drawn from the results need to be better discussed in the context of the importance of the placenta as another compartment with its own lipoprotein metabolism.

We thank the reviewer for their positive comments.

Abstract, line 42: To general and non-specific conclusion, authors should address what they mean with „...new molecular strategies...“ How would this work? Rather to conclude here based on their findings.

We have edited the abstract to include these suggestions (L31-48).

Discussion, line 427ff: The authors suggest that maternally derived LC-PUFAs in plasma are made available by phospholipids for the developing fetus in a LXR dependent mechanism. In the LXR DKO groups at late gestational there are FA-differences in the fetal liver (Figure 7L), but no changes of placental EPA and AA levels (Figure 7K). What is the role of the placenta in this concept?

- How do these phospholipids interact with the placenta?
- Uptake and transfer of those PUFAs to the fetus, which signals might be involved to end up in the fetal liver, but does have any effect in the placenta
- The proposed hepatic LC-PUFA-PL transcriptional changes as alternative to LXR-induced mechanisms nicely correlates with LC-PUFAs in the fetal liver, observed in this genetic context. Here again, the role of the placenta is not touched.

We agree that this is an important point, and we were initially surprised to find that maternal LXR α/β deletion (LXR-DKO) resulted in reduction in LC-PUFA accumulation in the fetal liver at 18.5 dpc without apparent alteration to the levels of these FAs in the placenta.

The transfer of LC-PUFAs from mother to fetal tissue must be considered as a multi-step process. Firstly, the FAs must be modified, packaged and mobilised by maternal tissues for transfer to the maternal plasma. Second, the FAs must be taken up by the trophoblast cells, potentially modified by placental metabolic enzymes, and transported across the placental barrier. Finally, the FAs must be transported into the fetal circulation, then incorporated into fetal tissues. The steady-state level of placental LC-PUFA levels will reflect several aspects of this process – maternal concentrations of circulating lipids, rate of uptake, accumulation into the placental metabolic pool and placental export.

The experiment using the LXR-DKO described in our manuscript functionally manipulates only the first part of this process – maternal supply of lipids to the fetus. Here we demonstrate clearly that LXR α/β function in the dam is necessary for ~50% of the total accumulation of fetal liver phospholipid (PL) associated DHA and arachidonic acid (ARA).

Without additional tracing experiments we cannot definitively say how the rate of placental transfer is modified in response to low maternal PL-LC-PUFA supply (stated as a limitation in our discussion). However, the lack of concordance between the placental and fetal compartment may be explained in several ways:

Firstly, we see higher overall concentration and a bigger differential between wild-type and LXR-DKO levels of plasma PL-DHA and AA at 14.5 dpc compared to 18.5 dpc (Figure 7J). This is consistent with the greater induction of expression of LXR target genes at the 14.5 dpc time-point (Figure 7B-H). We measured lipids in fetus and placenta at the end of gestation (18.5 dpc, earlier samples were not available). This stage represents the sum total of PL-DHA/AA accumulation in the fetus over the course of pregnancy, but not the steady state level in the placenta. Future work will employ stable isotope labelling methodologies to resolve this issue.

Secondly, recent studies employing mathematical models in combination with stable isotope tracing of FAs in a human placental perfusion system have demonstrated that the placenta

accumulates a relatively low level of DHA compared to other FAs (e.g, roughly 10-fold less compared to FA (16:0), PMID: 33637949). This is in agreement with historical findings that DHA is rapidly transported across the placenta with little placental accumulation. Interestingly an opposite effect is observed for n-6 PUFAs, particularly ARA which does accumulate in the placenta, possibly in reserve for prostanoid and endocannabinoid synthesis (PMID: 7277037, PMID: 32347309). In keeping with these findings, in our data PC-associated DHA in the placenta reached only 50% of the concentration in the fetal liver, replicating biomagnification (Figure 7K). Moreover, placental PC-ARA levels appeared to exceed those in the maternal plasma and fetal liver suggesting placental accumulation.

It has been shown that placentas from SGA (small gestational age) infants contained lower levels of omega-3 (ALA, EPA, DPA, and DHA) and high omega-6/omega-3 ratios (AA/DHA and LA/ALA), as well as low elongase (Elovl5) and high desaturase (D9Dn7 and D5Dn6) activity as compared to normal infants. How would such changes of PUFAs in the placenta, between different metabolic phenotypes in the mother, fit into your model?

How specific lipids and their FAs are partitioned to the placental metabolic pool or the fetal circulation is an extremely important topic, since these processes are modified by maternal diabetes and obesity (PMID: 37288271, PMID: 33637949). There is also an increasing appreciation that fetal/placental sex interacts with maternal diet to modulate placental adaptation to compromised maternal metabolic health (PMID: 32717842, 32452400 and see response to reviewer 2 below). Future studies where placental, fetal and maternal lipid metabolism are manipulated in isolation will be required to fully understand these important mechanisms.

Please add and discuss the role the placenta in context of your findings.

We have added the points detailed above to the discussion section (L566-591).

Although briefly touched in line 550ff differences between the lipoprotein metabolism between rodent and human, the authors should provide a in depth discussion paragraph on TGs in VLDLs between species and again to what extent is it feasible to translate these findings to human lipoprotein metabolism in pregnancy.

We agree that caution should be applied when extrapolating results from rodent pregnancy to other species, including humans. Differences between mouse and human lipoprotein metabolism have been underscored by many decades of work by investigators attempting to phenocopy the effects of cholesterol metabolism on cardiovascular disease. In this case it is clear that mice do not spontaneously generate atherosclerotic plaques, due at least in part by their low VLDL/LDL cholesterol levels compared to HDL-C, and lack of the enzyme CETP. In contrast to knowledge about cholesterol metabolism, much less is appreciated about mechanisms and species differences in lipid transport in the lipoprotein compartment. Although they are known to be 'HDL dominant', mice carry a significant proportion of their circulating phospholipids in LDL/VLDL (20-30% versus 50% in humans, PMID: 25894274).

While the widespread use of classical biochemical markers mean that levels of total cholesterol, bulk TG and HDL-C (and inferred measurements of LDL-C) are often reported in both human and rodent pregnancy studies, the reviewers will appreciate that details of the maternal lipidome are much sparser. While changes in the density, cholesterol and TG content have been described (PMID: 11134106), to our knowledge the FA composition in TG in different lipoprotein particle classes has not yet been fully described in pregnancy.

As the reviewer requests we have expanded the limitations section of the discussion to include some of these points, L613-625

Line487 ff: Role of the placental lipases and lysoPC symporter MFSD2A are discussed. How these well-known mechanisms for FA-uptake into the placenta are transferable to VLDL-mediated availability of triglycerides and LC-PUFA-PCs remains speculative. VLDL is hydrolysed by the hepatic lipase and its activity is regulated by PPARs. Moreover, studies indicate that HF/HCD-induced dysregulation of placental lipid hydrolysis contributes to fetal hepatic lipid accumulation and possibly to fetal overgrowth, at least in mice.

In this paper we show a generalised increase in circulating TGs in pregnancy. This rise is known to occur due to increased TG into all lipoprotein types (PMID: 34416270), but especially VLDL, due to reduced maternal tissue expression of lipoprotein lipase (LPL), hepatic lipase (HL) and increased expression of cholesterol ester transport protein (CETP). While rodents do not have CETP, the drop in adipose LPL is well reported (PMID: 5528598), and we saw a significant reduction in hepatic expression of HL (*Lipc*, Figure 5I). We believe that this increase in circulating TG in lipoproteins is the source of fatty acids of all types to the placenta. When incorporated into TG in VLDL, FAs are taken up into the placenta by a variety of mechanisms including VLDL-R and placental lipase-FATP-mediated routes.

Our data suggests that in addition to the elevation of all FAs in the maternal plasma (in TG), there is a specific, LXR-induced enrichment for LC-PUFAs in phospholipids. It is well known that endothelial lipase is the major lipase found on the maternal blood-facing surface of trophoblast cells in the human placenta (PMID: 17356047). EL is primarily a phospholipase A2 with high affinity for HDL, thus predicted to release *sn-1* FAs (PMID: 10192396). In contrast, lipoprotein lipase, the major TG lipase, is expressed at lower levels than EL in villous cytotrophoblasts and villous syncytiotrophoblast cells in early pregnancy and is not present at term (PMID: 17356047). Mice express high levels of both lipases in the labyrinthine zone of the placenta, but to our knowledge the cellular localisation has not been reported (PMID: 16150822). Interestingly, HDL in pregnancy is enriched for phospholipid LC-PUFAs compared to the other lipoproteins, suggesting a relatively unexplored role for HDL in gestational FA supply (PMID: 30670033). Overall, these data suggest that a major route for LC-PUFA transfer from lipoproteins is by EL-mediated cleavage of *sn-1* FAs from PLs, and potentially transfer of LPC-*sn2* lipids by MFSD2A, as noted previously by others and discussed by us in lines 503-524. We have expanded the discussion to clarify this point.

DLK1 (expression and loss of function) and localization in the human versus murine placenta should be addressed and discussed additionally.

DLK1 is highly expressed in the human and mouse placenta, and the expression pattern is very similar between species. Some ambiguity in the literature exists about this similarity, possibly because DLK1 is a secreted protein that may be detected in locations distal from its cellular source. This interpretation is based on substantial unpublished work from our group, including the use of high quality single-cell/nuclear sequencing datasets that have become available over the past few years (Data removed from reply).

In the mouse, *Dlk1* is expressed in the extraembryonic mesoderm-derived (EEM) cells of the placenta, and there is no contribution of mRNA from the trophoblast. We observe expression from midgestation (e9.5) in the allantoic cells invading the chorion, and subsequently in fetal endothelial cells and mesenchyme derived from the EEM (Figure 1a). This expression pattern is confirmed by single-nuclear sequencing of the mouse placenta from e9.5-e14.5 (PMID: 33141023), where *Dlk1* mRNA is confined to the same cell populations and is absent from the trophoblast (Figure 1b). We have previously reported DLK1 immunostaining in the mouse placenta where the *Dlk1* gene had been deleted from the EEM using conditional targeting (PMID: 27776119). In this case we identified a small number of putative trophoblast

cells which were positive for DLK1. We suggest that this protein may have been in transit across the trophoblast layer and was not directly synthesised by these cells, since we have never observed *Dlk1* in trophoblast using methods for mRNA detection.

In humans we see similar localisation of DLK1. In single cell data from the second trimester human placenta (GW17-24, PMID: 35796428) we can detect no *DLK1* transcripts produced from the trophoblast, instead there is very high expression in the EEM-derived mesenchyme of the villi, and a small amount in the fetal endothelial cells (Figure 1c). Similar to our work in the mouse, others have detected DLK1 protein in the syncytiotrophoblast layer (PMID: 26242929) – but to our knowledge mRNA expression has not been confirmed in these cells.

We propose that these data are not relevant to this manuscript. They will form part of a substantive paper about the role of DLK1 in placental function that we are currently preparing for publication. However, we have included more detail about the *Dlk1*-deletion model -please see our reply to Reviewer 3.

Reviewer #2 (Remarks to the Author):

In this study, in order to decipher the maternal adaptations required to provide optimal long chain polyunsaturated fatty acids (LC-PUFAs) to the developing fetus, and, in turn, prevent specific adverse neuronal infant outcomes, Risha Amarsi, Samuel Furse and coworkers conducted a very comprehensive study combining blood and liver lipidomics with liver transcriptional profiling on several independent genetically-murine models. The authors focus on the role of maternal hepatic lipids trafficking pathways in the lipid channel driven by the maternal-placental-fetal unit. This study is highly significant for better understanding of interactions between maternal lipid metabolism during gestation and adequate LC-PUFAS supplies to the conceptus.

The authors used virgin and pregnant mice models previously generated in their lab (8 models described in Figure 1A of genetically-mice with manipulated expression of DLK1) or generated by Nikolova and colleagues (LXR double knockout, DKO mice). Subsequent experiments were focused on dams' plasma and liver analyses of lipid classes (using untargeted DI-MS) and fatty acid composition of phospholipids (using targeted LC-MS/MS). Phospholipids were identified as candidate biomarkers using a stringent statistical pipeline combining multivariate (sparse PLS-DA) and univariate (FDR-corrected Student's t-test) workflow. Maternal hepatic biosynthesis and export of LC-PUFA-phospholipids were investigated by re-analyzing a microarray dataset combined with a targeted ChIP enrichment analysis (ChEA) and a validation of genes of interests using RT-qPCR. The genes of interests were divided into genes involved in limiting steps (PUFA desaturation, PC biosynthesis, lipoprotein assembly), in Kennedy pathway, reverse cholesterol transport and de novo lipogenesis.

Overall, the manuscript represents a set of well conducted experiments that provide novel insights into the complex interplay between maternal hepatic lipid metabolism and the biomagnification of the essential PUFA towards the placental-fetal unit. The key results of this study relate to the activation, in late pregnancy, of LC-PUFAs synthesis, from maternal diet precursors, and their incorporation into phospholipids through a specific transcriptional program regulated by their liver X receptor (LXR) and modulated by maternal production of the Delta-like homologue 1 (DLK1), a major protein of the conceptus, previously identified by the authors, as a putative key regulator of maternal fatty acid metabolism during gestation. However, I have some minor issues with some of the data and discussion of the literature

that should be addressed before publication:

Major comment:

1- In the introduction section, the paragraph (lines 64-72) describing PUFA can be considerably shortened however, it would have been more interesting to specify why LC-PUFAs are crucial for central nervous system development across the last trimester of pregnancy and early postnatal months as LCPUFAs are considered essential in fetal life due to low or absent delta 5 and 6 desaturase activity in the placenta and fetus. Moreover, there is no elements concerning the sexual dimorphism in specific adverse neuronal infant outcomes due to low n-3 PUFA supplies, such as males who were reported to be more likely to develop attention deficit/hyperactivity in pregnancy with obesity context.

We have made some edits to this section to prioritise the information (L70-97). We agree that there is strong evidence for low and absent delta 5 and 6 desaturase activity in the fetus (we cite PMID: 7285840 in the manuscript). However, there is some evidence supporting placental production of these enzymes in rodents and humans (for example PMID: 32717842, 32452400). For this reason we have not modified the sentence “Fetal synthesis can only account for a small proportion of this demand, and instead maternal production/mobilisation and placental transfer is required to meet the ARA and DHA requirements for healthy development” (L79-81).

2- Why the sex factor was not considered in the present study due to the putative flexibility in conceptus-maternal interactions that not only could impact secretory factors derived from either the conceptus or endometrium and also could be altered by conceptus sex? The authors could also develop this point in the discussion section.

We are aware of the important fetal sex differences in both placental responses to LC-PUFA uptake and developmental outcome in response to maternal obesity described by Powell et. al. and others (PMID: 33321178, PMID: 37770949). However, our study focuses primarily on the maternal hepatic mechanism of biomagnification rather than the fetal/placental uptake route. Because rodent pregnancies contain multiple pups of mixed sex, it is not possible to dissect the impact of fetal sex on the maternal hepatic response in our study (as it is likely to be an average of both sexes). Our future work can address both placental compensatory mechanisms of reduced maternal PUFA supply (as mentioned in the response to Reviewer 1, above) and should include offspring sex in this analysis. Consequently, this is outside the scope of the current work, which is already a complex story, and so we have not included these points in the Introduction. However, since the issue of fetal sex is an important one, we have included the following sentence in the limitations section of the Discussion; “Recent work has highlighted an important role for fetal sex in placental metabolism and transfer of DHA in the context of maternal obesity. Specifically, male offspring of obese mothers have reduced placental and plasma levels of DHA compared to females, or the offspring of lean mothers (PMID: 33321178, PMID: 37770949). Our study focuses on the maternal hepatic mechanism of biomagnification rather than the fetal/placental uptake route. Because rodent pregnancies contain multiple pups of mixed sex, it is not possible to dissect the impact of fetal sex on the maternal hepatic response. Future work must address both placental compensatory mechanisms of reduced maternal LC-PUFA supply and should include offspring sex in this analysis.” (L604-612).

3- In the results section, lines 140-143, have the authors any other information about the mitochondrial function such as beta oxidation which can be accessed through the acylcarnitine profiles of dam’s plasma?

We did not measure acylcarnitine profiles in these mice, as this class of metabolites is not covered by the lipidomics method. Since the animals were free-fed we did not expect to observe an appreciable level of mitochondrial beta-oxidation. Nevertheless, as part of our bioactive lipids panel we measured hepatic 3-hydroxy-medium and long-chain fatty acids. We observed no differences between groups (Figure 4A).

4- In the results section, lines 320-327, the authors can also investigate directly the phospholipase A2 activity through the ratio Lyso-PC/PC.

We attempted to use the LPC(16:0)/PC(36:4) and LPC(18:0)/PC(38:4) ratios in the liver as a proxy to determine if PLA₂ activity was increased in pregnancy. We saw the opposite relationship. We suggest that this signal was swamped by the large increase in LC-PUFA synthesis at this time, making the PLA₂ effect hard to isolate. A good estimation of PLA₂ activity must assume similar rates of synthesis between conditions, which is not the case in our data.

6- In the discussion section, a putative cross-talk between maternal liver and the microvillous membrane of the syncytiotrophoblast of human placental and its putative involvement in the gradient of LC-PUFA supply to the placenta-fetal unit could be raised, due to the recent literature about fetal sex-dependent placental adaptations in pregnancy complicated by obesity, or diabetes. Indeed fatty acid transfer proteins or enzymes involved in LCPUFA synthesis can be differentially altered regarding hepatic or placental compartment.

We have expanded the Discussion section to include a section about the complexity of the interplay between maternal, fetal and placental compartments, including the role of fetal sex in placental adaptations to compromised maternal metabolism; obesity and diabetes (L566-593).

5-Lines 461-462, the authors could also consider the sphingomyelin data in their untargeted lipidomics analysis, as there are also choline donors such as PC- Their findings are also in adequation with the higher efficiency of DHA, provided as a phospholipid, than DHA supplied as part of a triglyceride, for brain DHA accretion and infant growth, as reported in literature.

The sphingomyelin (SM) data is shown in Tables S1 and S2. Since SMs were not altered in pregnancy as a class, and only 2 SMs were found to show differential abundance in our candidate biomarker analysis (Table S4), SM(36:1) (up in pregnant liver) and SM(33:1) (up in pregnant plasma), we have chosen not to highlight these results in an already complex paper.

We have not included the second point as it appears to be unresolved in the literature. Indeed, a recent substantive review of fetal DHA accretion according to lipid source stated that TGs and PLs perform equally well with current levels of supplementation, but concluded that more data is required to conclusively determine this point (PMID: 33557158).

6-Lines 504-512, the high levels in hepatic ARA-enriched phospholipids raises the role of oxylipins and COX/LOX pathways in pregnancy, and notably pregnancy complicated by obesity, preeclampsia...Could the authors discuss this point and have they arguments for specific inflammatory environment associated to their genetically mice models?

We agree that the contribution of prostanoid synthesis to pregnancy complications is an exciting topic that is highlighted by our analysis. To our knowledge ours is the first unbiased screen of hepatic bioactive lipids in pregnancy, so much work will need to be done to follow up these initial findings. Our animals are housed in a 'clean' but not sterile facility, thus we do

not expect the animals to be exposed to a specific inflammatory environment. Since we feel it is too soon to speculate about the relationship between specific COX-derived lipid mediators and pregnancy complications, we wish to keep the current wording of this section as “Given the reported associations between prostanoids and pre-eclampsia (70–72), and their critical regenerative roles following hepatic injury (73), further research into maternal hepatic bioactive PUFA metabolites may elucidate underlying mechanisms that implicate liver dysfunctions to pregnancy complications. “ (L534-538).

The figures and Tables are very clear and the legends provide full of details. We appreciate the study design and schematic depicting lipid biosynthesis pathways and/or the transcriptomic analysis.

We thank the reviewer for their extremely helpful comments throughout.

1- In the title, please develop LC-PUFA

This has been done (L2).

2- P2-. In the abstract section, replace “untargeted lipidomics” by “a comprehensive approach combining untargeted and targeted lipidomics with” and specify which “genetically-manipulated murine models” were used.

We have edited the abstract to include these suggestions (L31-48).

3- In the Methods section, lines 770-777, the authors reported animal study in accordance with the ARRIVE (Animal Research: Reporting of In Vivo Experiments) guidelines. Please add the details of energy supply for each macro-nutrients in the standard chow diet (Supplementary Table S12).

These values have been added to Table S13.

4- In Supplementary Table S6, undetermined means under LOD ??

Yes, the table legend has been amended to make this clear.

5- Concerning the sentence lines 788_790, “Age-matched WT (group 1), Mat (group 6) and null (group 3) females were used as virgin control groups”, it would be interesting to know if pregnant females presented similar pre-gravid body weight as those of virgin females?

Body mass did not differ between groups 1, 6 and 3. However, adipose mass was significantly increased in group 3 (abdominal white adipose tissue mass = 0.28g in group 3 compared to 0.18g in group 1 and 0.17g in group 6). This is detailed in our previous paper (PMID: 27776119). We have modified the discussion to more clearly articulate this. (L546-551)

6- Lines 982, please develop the abbreviations Hprt and Tbp

This has been done.

7- In the statistical section, lines 1028-1031, it would be appreciable to detail the different tests used to compare the groups as reported in figures' legends.

We have added: In general, two-tailed Student's t-tests were used to perform pairwise comparisons, whereas One-Way ANOVA was used to compare multiple groups, as detailed in respective Figure legends. (L1110-1112).

8- Line 1032 Please add the agreement of the animal study in the study approval section

This has been added.

Reviewer #3 (Remarks to the Author):

This study extends the authors' previous study (2016 Cleaton MAM et al.), and the authors conducted lipidomics, PUFA metabolite analysis, and Transcriptional analysis using stored samples derived from various cohorts of mouse models with different genotypes. The authors demonstrated a late pregnancy-specific, selective activation of the Liver X Receptor signaling pathway which dramatically increases maternal supply of LC-PUFAs within circulating phospholipids. Although this study applied new analytical methods and showed interesting findings, the reviewers have several comments and some concerns.

Major comment 1

The authors previously stated that Fetus-derived DLK1 is required for maternal metabolic adaptations to pregnancy (2006 Cleaton MM et al.). However, the present study showed that the hepatic transcriptional program is co-ordinately regulated by the liver X receptor (LXR) and modulated by maternal production of DLK1. The reviewer could not understand how to interpret the conclusions from the previous study and the findings in the present study. Which is critical DLK1 from the mother or fetus? The reviewer considers that a more careful explanation is needed.

We hope that the concerns expressed in this comment are dealt with by the changes described below, and in modifications suggested by the other reviewers.

Major comment 2

The authors described as follows; We observed that loss of DLK1 in the fetus caused impairments in maternal fasting metabolism and lipoprotein production, whereas loss of DLK1 in the dam prevented the normal acquisition and release of her adipose tissue stores (lines 59-62). The authors need to describe what is known and what is unknown about how DLK1 affects fatty acid metabolism in pregnant mothers. Furthermore, the authors need to show the known information in the introduction section on what kind of fatty acids are affected by DLK1 in pregnant females.

Prior to this work the unbiased lipid profile of the pregnant mouse in late gestation had not (to our knowledge) been reported. In addition, we had no information about changes to the lipid species associated with DLK1 modification in pregnancy. In the Cleaton 2016 (PMID: 27776119) paper we had described the lipid pathways in only broad brushstrokes – measuring serum cholesterol (including HDL-cholesterol), TAG, unesterified fatty acids and a major ketone species (3-hydroxy-butyrate). In addition we had measured maternal weight gain and body composition.

To make it clear to the reader that this was our starting point, we have modified the section highlighted by the reviewer: “This work led us to hypothesise that DLK1 is a key modulator of maternal fatty acid metabolism in pregnancy, yet a detailed examination of pregnancy-

associated lipid species has not been performed.” (L67-69).

Major comment 3

The mother may not produce DLK1 but the fetus may have DLK1 production (G5), or the mother may produce DLK1 but the fetus may produce DLK1 (G7). Reviewers would like to know whether the total amount of DLK1 in late pregnancy is important or whether DLK1 production differs depending on whether it originates from the mother or fetus.

In adult female mice plasma DLK1 levels are low (<60ng/mL). However, in late gestation these levels rise 5x to 250-300ng/mL in the maternal plasma. Fetal DLK1 is the source of this DLK1 in the maternal circulation (PMID: 27776119). In terms of the sum maternal plasma DLK1 in pregnancy, the major contribution is thus of fetal, rather than maternal tissue origin. To clarify this point we have added the previously measured mean plasma DLK1 levels by experimental group to the schematic in Figure 1A.

However, in this study we found that it is the maternal genotype that contributes most to the maternal mobilisation of PUFAs. To clarify further we added the following sentence to the Discussion: “In our previous study we found that fetally-derived DLK1 in maternal circulation had a minimal impact on fed metabolism (4). Consistently, here fetal DLK1 led to a small shift in total PE:PC ratios (Figure 2I, J). However, the ablation of *Dlk1* in maternal tissues modified the dynamics of lipid mobilisation, potentially via indirect effects on whole body adipose storage.”

Major comment 4

The authors compared various lipids fatty acids levels and transcriptional levels of targeted genes in G3 and G5 as well as in G1 and G2 and in G6 and G8. G3 and G5 have no DLK1 from maternal own origin, while G5 has circulating DLK1 from fetal origin. The reviewer would like to know whether the circulating DLK1 levels in G3 and G5 are the same or different from those in G1, G2, G6, and G8.

Please see the comment above.

Major comment 5

The authors compare maternal circulating and hepatic lipid profiles among the mice models with different maternal genotypes WT, Mat (DLK1+), and Null (DLK1-) between non-pregnancy and pregnancy, respectively (Figures 2A-H). Similar changes are observed in each genotype between non-pregnancy and pregnancy, but TG (Liver) is not different between non-pregnancy and pregnancy in the null (DLK1-) mice model (Figure 2B). Does the absence of DLK1 derived from the mother have anything to do with this?

We agree, and have modified part of the discussion “We found that dams lacking DLK1 derived from maternal tissues had modestly increased transcription of rate-limiting steps in the hepatic synthesis and export of LC-PUFA-phospholipids. Notably, in our previous study, these mice commenced pregnancy with a higher adipose mass, but exhibited a reduced ability to gain adipose tissue during pregnancy (PMID: 27776119), and appear to accumulate less hepatic TG (Figure 2B).” L546-551.

Major comment 6

The authors described as following: Despite the influence of DLK1 on hepatic PE(18:0/20:4), no effect was observed on PUFA metabolite concentration (lines 264-265). Could you show the related data?

All data from the PUFA metabolite screen is now shown in Supplementary Table S8.

Minor comment 1

In line 27, the authors cited Figure 3B. Is it a mistake of Figure 5B?

Changed, thanks.